# The Multifaceted Role of p53 in Cancer Molecular Biology: Insights for Precision Diagnosis and Therapeutic Breakthroughs

**DOI:** 10.3390/biom15081088

**Published:** 2025-07-27

**Authors:** Bolong Xu, Ayitila Maimaitijiang, Dawuti Nuerbiyamu, Zhengding Su, Wenfang Li

**Affiliations:** 1School of Pharmaceutical Science, Institute of Materia Medica, Xinjiang University, Urumqi 830017, China; xubolong@stu.xju.edu.cn (B.X.); ayitila@xju.edu.cn (A.M.); nuerbiyamu@stu.xju.edu.cn (D.N.); 2College of Life Science and Technology, Xinjiang University, Urumqi 830017, China

**Keywords:** p53, cancer, molecular biology, precision diagnosis, therapeutic breakthroughs

## Abstract

The protein p53, often referred to as the “guardian of the genome,” is essential for preserving cellular balance and preventing cancerous transformations. As one of the most commonly altered genes in human cancers, its impaired function is associated with tumor initiation, development, and resistance to treatment. Exploring the diverse roles of p53, which include regulating the cell cycle, repairing DNA, inducing apoptosis, reprogramming metabolism, and modulating immunity, provides valuable insights into cancer mechanisms and potential treatments. This review integrates recent findings on p53′s dual nature, functioning as both a tumor suppressor and an oncogenic promoter, depending on the context. Wild-type p53 suppresses tumors by inducing cell cycle arrest or apoptosis in response to genotoxic stress, while mutated variants often lose these functions or gain novel pro-oncogenic activities. Emerging evidence highlights p53′s involvement in non-canonical pathways, such as regulating tumor microenvironment interactions, metabolic flexibility, and immune evasion mechanisms. For instance, p53 modulates immune checkpoint expression and influences the efficacy of immunotherapies, including PD-1/PD-L1 blockade. Furthermore, advancements in precision diagnostics, such as liquid biopsy-based detection of p53 mutations and AI-driven bioinformatics tools, enable early cancer identification and stratification of patients likely to benefit from targeted therapies. Therapeutic strategies targeting p53 pathways are rapidly evolving. Small molecules restoring wild-type p53 activity or disrupting mutant p53 interactions, such as APR-246 and MDM2 inhibitors, show promise in clinical trials. Combination approaches integrating gene editing with synthetic lethal strategies aim to exploit p53-dependent vulnerabilities. Additionally, leveraging p53′s immunomodulatory effects through vaccine development or adjuvants may enhance immunotherapy responses. In conclusion, deciphering p53′s complex biology underscores its unparalleled potential as a biomarker and therapeutic target. Integrating multi-omics analyses, functional genomic screens, and real-world clinical data will accelerate the translation of p53-focused research into precision oncology breakthroughs, ultimately improving patient outcomes.

## 1. Introduction

Tumor suppressor p53 is a cornerstone of cancer molecular biology and is renowned for its unparalleled role in safeguarding genomic integrity and preventing oncogenic transformations. Over four decades ago, p53 evolved from an enigmatic cellular protein to a central regulator of cellular stress responses, earning its designation as the “guardian of the genome” [1]. The importance of this gene is highlighted by its prevalence as the most commonly mutated gene in human cancers, with somatic alterations observed in more than 50% of malignancies, such as lung, breast, colorectal, and ovarian cancers [2]. These mutations eliminate the tumor-suppressing capabilities of p53 and introduce new gain-of-function (GOF) properties that enhance tumor progression, resistance to therapy, and metastatic potential [3].

At the molecular level, p53 functions as a transcription factor that orchestrates a diverse array of cellular processes including cell cycle arrest, DNA repair, apoptosis, senescence, and metabolic reprogramming [4]. Its activity is tightly regulated by post-translational modifications, protein–protein interactions, and subcellular localization, ensuring precise responses to genotoxic, oxidative, or oncogenic stressors. Wild-type p53 acts as a molecular sentinel, halting cell cycle progression to allow DNA repair or triggering programmed cell death when damage is irreparable. Conversely, mutant p53 proteins frequently accumulate in the nucleus because of impaired degradation, disruption of normal p53 signaling, and engagement in aberrant transcriptional programs that promote cell survival, proliferation, and immune evasion [5]. In addition to its classical roles, p53 has emerged as a key regulator of tumor microenvironment (TME) dynamics and immune responses. Recent studies have revealed its involvement in modulating immune checkpoint expression, cytokine secretion, and antigen presentation, thereby influencing the efficacy of immunotherapies, such as anti-PD-1/PD-L1 agents [6]. This immunomodulatory capacity positions p53 as a critical link between intrinsic tumor biology and extrinsic host defense, highlighting its potential to guide personalized therapeutic strategies. Clinical translation of p53 research has been hampered by the complexity of its signaling network and the heterogeneity of its mutations. While some p53 mutations are “hotspots” (e.g., R175H, R248Q, R273H) with well-characterized GOF effects, others exhibit context-dependent activities, complicating therapeutic targeting [7]. However, advances in precision diagnostics, such as liquid biopsy technologies for non-invasive mutation detection and machine learning algorithms for predicting p53 functional status, have enabled earlier cancer detection and patient stratification [8,9]. Concurrently, the development of small molecules capable of restoring wild-type p53 activity (e.g., APR-246) or destabilizing mutant p53 proteins (e.g., MDM2 inhibitors) has spurred renewed interest in p53-based therapies [10].

This review synthesizes recent breakthroughs in understanding p53’s multifaceted roles in cancer biology, with a focus on its implications for precision diagnosis and therapeutic innovation. By integrating insights from molecular mechanisms, diagnostic tools, and emerging clinical trials, we aimed to highlight the transformative potential of p53-targeted strategies in oncology.

## 2. Molecular Mechanisms of p53

### 2.1. Transcriptional Regulation

Transcriptional regulation of p53 has been extensively studied in recent decades. p53, as an important transcription factor, suppresses tumors through sequence-specific DNA binding to p53 response elements (p53REs) [11,12,13,14]. This well-established mechanism involves the activation of genes that control cell cycle arrest (CDKN1A/p21), DNA repair (GADD45 family), apoptosis (BAX and PUMA), senescence (CDKN2A/p16INK4a), and metabolic reprogramming [15,16]. The tumor suppressor p53 functions as a central regulator of cellular stress responses through its multifaceted transcriptional regulatory network, and acts as a critical determinant of genomic stability and tumor suppression [17,18].

The p53 activity equilibrium is dynamically modulated by antagonistic regulatory axes [19]. The MDM2-MDM4 heterodimer constitutes the primary negative regulatory circuit, with MDM2 functioning as an E3 ubiquitin ligase that catalyzes lysine 48-linked polyubiquitination at multiple sites (K48 residues) to direct p53 proteasomal degradation [20,21]. This degradation pathway is counteracted by the ARF tumor suppressor, which sequesters MDM2 in the nucleolar compartments through direct interactions, thereby preventing p53 ubiquitination and promoting its nuclear accumulation [22]. The PI3K–AKT survival signaling axis further refines this balance by phosphorylating MDM2 at the S166/S186 residues, enhancing its nuclear import efficiency and E3 ligase activity while simultaneously suppressing histone deacetylases (HDACs) to facilitate p53 acetylation at K382, a modification critical for transcriptional activation [23]. Covalent modifications to the N-terminal transactivation domain (TAD) and DNA-binding domain (DBD) form a combined regulatory code [24]. DNA damage-induced phosphorylation (e.g., ATM/ATR-mediated S15 and S37) disrupts MDM2 binding while creating docking sites for bromodomain-containing readers (e.g., BRD4) [25]. Concurrently, p300/(CAMP-response element binding protein) CBP-catalyzed acetylation at K382 stabilizes p53–DNA interactions and recruits chromatin-remodeling complexes [26]. DBD undergoes allosteric regulation under oncogenic stress, adopting conformations that promote phase separation into membraneless compartments, thereby concentrating the transcriptional machinery at super-enhancers associated with pro-apoptotic targets [27,28].

Genotoxic insults trigger p53 dimerization and initiate transcriptional cascades that influence cell fate decisions [29,30]. Double-strand breaks activate the ATM/ATR–CHK1/2 axis, thus leading to p53 stabilization-mediated G1/S arrest and DNA repair [31]. Metabolic stress signals (e.g., AMPK activation) induce phosphorylation at S46, shifting the transcriptional output from cell cycle arrest to pro-apoptotic programs via ASPP1 coactivator recruitment [32]. Noncoding RNAs (LINC01629, LINC01164, and Wrap53) and chromatin modifiers (SWI/SNF) play important roles in the p53 regulatory network [33,34,35,36]. The robustness of this system is underscored by its integration with oncogenic signaling; activated RAS induces replicative stress (activating p53) and upregulates MDM2, creating a failsafe mechanism to limit neoplastic transformation [37].

This intricate interplay of regulatory layers, including protein degradation, post-translational modification networks, liquid–liquid phase separation, and metabolic adaptation, positions p53 as a sentinel node for maintaining genomic integrity. Dysregulation of these mechanisms contributes to more than 50% of human cancers, highlighting the therapeutic potential of targeting p53 regulatory nodes in oncology.

### 2.2. Post-Translational Modifications

p53 undergoes several key post-translational modifications, including phosphorylation, acetylation, and ubiquitination. These modifications play critical roles in the regulation of p53 function, particularly its capacity to regulate autophagy.

Acetylation constitutes a key regulatory switch in p53 post-translational modification networks that dynamically controls its functional outcomes through site-specific lysine modifications. The sirtuin (SIRT1)/SLC7A11 signaling pathway is a classical signaling pathway involved in iron deficiency [38,39]. Emerging evidence suggests that SIRT1, a class III histone deacetylase, retards apoptosis and reduces oxidative stress-induced damage by deacetylating p53 [40]. The transcriptional activity of genes inhibited by p53 deacetylation activates the transcription of the downstream gene SLC7A11, inhibits ROS accumulation and response, and ultimately inhibits iron death [41]. p300/CBP-mediated acetylation at lysine 382 within the C-terminal regulatory domain (CTD) enhances DNA-binding affinity by neutralizing positive charges, while also serving as a recognition motif for TIP60 acetyltransferase that activates pro-apoptotic target genes [42].

Phosphorylation can alter the stability of p53, thereby regulating its half-life in cells [43]. During transcription, TAF1 phosphorylates TAF7 through its Ser/Thr domain, which dissociates phosphorylated TAF7 from the TF II D complex and enhances transcriptional activity [44]. TAF1 can downregulate the expression of the tumor suppressor p53 by phosphorylating p53 at Thr55 or promoting the expression of the oncogene MDM2, ultimately inducing cellular G_1_/S-phase progression, which is involved in the regulation of the cell cycle [45,46,47]. When the cell receives an external stimulus, the p53 Ser15 and 20 sites are modified by phosphorylation, which blocks the interaction between MDM2 and p53, causing the protein level of p53 to increase and activate the transcriptional activity of p53, which controls the fate of the cell by regulating the relevant target genes [48]. DNA damage-induced ATM/ATR kinases phosphorylate p53 at the serine 15 and serine 37 residues, disrupting MDM2 binding and enabling rapid p53 stabilization [25].

The most common example of ubiquitination is the MDM2–p53 regulatory axis. MDM2 binds to the transcriptional activity domain of p53, promoting the ubiquitination and subsequent degradation of p53, thereby disrupting the genomic protective function of p53 [49,50]. This regulatory function has been identified as a promising target for the development of anticancer drugs. Nutlins are a class of cis-imidazole small-molecule MDM2-p53 antagonists identified by screening synthetic chemical libraries, represented by the compound nutlin-3 [51]. Nutlin-3 disrupts the interaction between p53 and MDM2, inhibits the ubiquitination and degradation of p53 by MDM2, stabilizes the p53 state, improves the expression of p53 in wt p53 cells, and induces the p53 signaling pathway to exert anti-tumor effects [52].

Post-translational modifications (PTMs) operate as combinatorial codes that finely tune the p53 functional output [53]. As shown in Figure 1, these covalent modifications integrate diverse cellular signals to fine-tune p53 stability, transcriptional output, and non-genomic functions. Critically, the deregulation of PTM networks observed in over 50% of human cancers creates unique therapeutic vulnerabilities [54,55]. We posit that mapping patient-specific PTM signatures will enable the precise targeting of p53 dysregulation, as cancers with SIRT1 overexpression may benefit from acetylation-focused therapies. Realizing this precision medicine paradigm requires the development of modulators of PTM writers/erasers, and diagnostic tools to decode the p53 modification profiles of individual patients.

### 2.3. Non-Transcriptional Functions

The tumor suppressor p53 exerts its pro-apoptotic functions through a multifaceted array of non-transcriptional mechanisms that operate independently of its canonical DNA-binding activity [56]. Beyond its role as a transcription factor, p53 has been shown to localize to mitochondrial membranes, where it directly engages BCL-2 family proteins to orchestrate mitochondrial outer membrane permeabilization (MOMP) [57]. In this subcellular compartment, p53 physically interacts with pro-apoptotic BAX/BAK oligomers, stabilizing their pore-forming conformations, and facilitating cytochrome c release into the cytosol [58]. This interaction network is further modulated by competitive binding to anti-apoptotic MCL1, where p53 disrupts MCL1′s sequestration of BH3-only proteins, thereby liberating pro-apoptotic effectors, such as BIM and tBID [59].

The non-transcriptional apoptotic program is amplified through p53′s direct engagement with BH3-only protein modules [60]. p53 forms ternary complexes with PUMA/NOXA and BCL-XL/BCL-2, neutralizing the anti-apoptotic activity of these survival factors, while simultaneously promoting BAX activation. This dual regulatory mechanism enables p53 to bypass transcriptional delays and execute rapid apoptotic responses in the presence of severe DNA damage or irreparable genomic stress [61]. The mitochondrial localization of p53 is dynamically regulated by post-translational modifications, with phosphorylation events at S392 and S46 enhancing its affinity for cardiolipin-rich membrane domains [62]. Previous structural studies have revealed that p53 employs distinct protein-interaction interfaces because of its non-transcriptional functions [63]. While the DNA-binding domain (DBD) remains critical for transcriptional regulation, the tetramerization domain and C-terminal regulatory region mediate direct interactions with BCL-2 family members [64]. This functional dichotomy allows p53 to integrate apoptotic signals through parallel pathways; transcriptional induction of PUMA/BAX combined with direct mitochondrial activation creates a failsafe mechanism to ensure cell death [65]. The physiological significance of these non-transcriptional mechanisms is underscored by their conservation across evolutionary contexts. Similarly, during metabolic crises, p53 maintains energy homeostasis by selectively eliminating damaged organelles in an AMPK-dependent manner [66].

This non-canonical apoptotic pathway intersects with other cell death pathways, including necroptosis and ferroptosis, through the p53-mediated modulation of lipid metabolism and iron homeostasis [67]. The system’s robustness is further enhanced by crosstalk with innate immune signaling, where mitochondrial p53 promotes inflammasome activation through release of damage-associated molecular patterns (DAMPs) [68].

Collectively, these non-transcriptional functions have established p53 as a multifunctional guardian that executes context-dependent cell fate decisions through the direct integration of subcellular signaling networks, as shown in Figure 2.

## 3. p53 in Tumor Microenvironment and Immune Modulation

### 3.1. Angiogenesis Regulation

Angiogenesis is a complex and critical process that facilitates tumor growth, invasion, and metastasis to nearby tissues [69]. p53 is a key regulator of angiogenesis and functions as a negative modulator through various mechanisms. [70]. Tumor dysfunction promotes angiogenesis. It is commonly acknowledged that p53 can interfere with angiogenesis through three distinct mechanisms. First, p53 has been found to suppress the hypoxia-sensing system by interacting with the hypoxia-inducible factor (HIF) signaling pathway [70]. One of the most significant transcription factors implicated in the context of tumor hypoxia is HIF-1α, which has been demonstrated to exert a pivotal role in regard to modulating the expression levels of vascular endothelial growth factor (VEGF). HIF-1α activity is inhibited by p53, and p53 has been shown to directly bind to HIF-1α. It has been demonstrated that HIF-1α activity can be inhibited by p53, which has been shown to bind directly to HIF-1α and target the protein for degradation. In addition, evidence has been presented which suggests that this is the wild-type [71]. Second, p53 directly affects the angiogenic process through transcriptional repression of pro-angiogenic factors. It has been established that p53 exerts its tumor-suppressing effects by repressing four genes that encode proangiogenic factors, including VEGF, basic fibroblast growth factor (bFGF) [72], bFGF-binding protein, and cyclooxygenase-2 (COX-2). Li et al. indicated a potential role for mutant p53 in angiogenesis and metastasis by upregulating VEGF expression in patients with pancreatic cancer [73]. The observed inhibition of COX-2 was attributed to a mechanism in which p53 competes with TATA box-binding proteins for binding to the COX-2 promoter. This process is believed to play a pivotal role in suppressing angiogenesis [74]. Several past studies have demonstrated that COX-2 exhibits an indirect increase in VEGF expression and a concurrent positive correlation with angiogenesis in numerous cancers [75]. In cancer cells, the absence of wild-type p53 increases COX-2 and angiogenic factor expression [76]. Concurrently, vasopressors, including platelet-responsive protein-1, are inhibited, which inducing angiogenesis [77]. This suggests that p53 inhibits crucial pathways involved in prostaglandin-mediated angiogenesis.

It has been demonstrated that wild-type p53 exerts its inhibitory effect on angiogenesis through the increased expression of vascular inhibitory factors. Furthermore, p53 has been shown to upregulate a variety of different genes that inhibit angiogenesis, including platelet-responsive protein-1 (TSP-1) [78], brain-specific inhibitor of angiogenesis 1 (BAI1) [79], hepatic ligand protein receptor A2 (EPHA2) [80], and anti-angiogenic collagen [81].

In general, p53 acts as a negative regulator of tumor angiogenesis through multifaceted mechanisms. Consequently, its frequent dysfunction in cancers eliminates this critical suppression, enabling uncontrolled blood vessel formation that facilitates tumor growth and spread.

### 3.2. Immune Surveillance

Tumor suppressor p53 functions as a central regulator of immune surveillance within the tumor microenvironment (TME) by integrating the cellular stress responses with innate and adaptive immune activation. Beyond its canonical role in maintaining genomic stability, p53 directly modulates tumor cell immunogenicity through multiple parallel pathways [82]. By transcriptionally upregulating MHC class I molecules and antigen-processing machinery components (TAP1 and ERAP1), p53 enhances tumor antigen presentation and facilitates cytotoxic T lymphocyte (CTL) recognition [83]. This immunostimulatory effect is augmented by p53-dependent secretion of interferon-β (IFNB1), which activates dendritic cell (DC) maturation and NK cell cytotoxicity through paracrine signaling [84]. At the interface of innate immunity, p53 promotes NK cell-mediated tumor clearance via induction of stress ligands ULBP1/2 on tumor cells, creating an “eat-me” signal for activating receptors (NKG2D) [85]. Concurrently, p53 suppresses immunosuppressive pathways by downregulating CD47 (“don’t-eat-me” signal) and IDO1 (tryptophan catabolism), thereby relieving macrophage polarization towards the pro-tumoral M2 phenotype [86]. The miR-34 family, directly induced by p53, forms a regulatory axis with PD-L1 expression, creating a negative feedback loop that modulates CTL exhaustion [87]. The immunomodulatory functions of p53 extend to TME remodeling through metabolic reprogramming. Wild-type p53 suppresses aerobic glycolysis (the Warburg effect) by inhibiting GLUT1/3 and PFKFB3, and reducing lactate accumulation, which otherwise impairs CTL effector functions [88]. Conversely, the mutant p53 isoforms promote immunosuppressive metabolome shifts by upregulating ARG1 and IDO2 in myeloid-derived suppressor cells (MDSCs), while simultaneously enhancing VEGF secretion to establish pro-angiogenic T cell-exclusionary niches [89,90].

Structural variations of p53 differentially affect immune surveillance. GOF mutants exhibit enhanced binding to the PD-L1 promoter, creating an immunoevasive phenotype that correlates with resistance to anti-PD-1 therapy [91]. In contrast, temperature-sensitive p53 mutants retain partial immunostimulatory capacity and maintain MHC-I expression, while failing to induce apoptosis, creating a unique vulnerability to combined chemoimmunotherapy methods [92]. This intricate interplay between p53 status and TME composition underscores its dual role as both an immune activator and potential suppressor [93]. Therapeutic strategies targeting p53-dependent immunomodulatory pathways, including STING agonists that synergize with p53 reactivation or miR-34 mimetics to restore immune checkpoint control, are emerging as promising avenues to overcome treatment resistance in p53-deficient malignancies [94]. Therefore, p53 is important for preserving immune surveillance by regulating cellular stress responses, facilitating signals that amplify immune responses, and helping the immune system recognize and destroy aberrant cells (Figure 3).

### 3.3. Inflammation Within the Tumor Microenvironment

p53 is a critical modulator of inflammatory dynamics within the TME and plays a role in the complex crosstalk between malignant cells, stromal components, and immune effectors [95]. The tumor microenvironment (TME) encompasses non-cancerous cells and surrounding cellular elements, such as immune cells, blood vessels, stromal cells, extracellular matrix, and lysogenetic factors [96]. Wild-type p53 functions as a brake in protumorigenic inflammation via multiple parallel mechanisms. Inflammation in the tumor microenvironment facilitates the development of malignant processes such as epithelial–mesenchymal transition (EMT), metastasis, and angiogenesis [97]. In addition, the mutation or loss of function of the p53 protein in epithelial cells is important for maintaining a chronic inflammatory state by regulating different molecular regulators [98]. From a mechanistic perspective, p53 inactivation has been shown to deregulate the inhibition of the NF-κB signaling pathway in activated B cells [99]. This deregulation produces inflammatory cytokines that exert the ability to induce inflammation in epithelial cells.

By suppressing NF-kB transcriptional activity via direct protein–protein interactions and competitive binding to coactivators like CBP/p300, p53 attenuates the expression of proinflammatory cytokines including IL-6, IL-8, and TNF-α [99]. This inhibitory effect extends to myeloid-derived suppressor cell (MDSC) recruitment, as p53-deficient tumors exhibit elevated CCL2/CCL5 chemokine secretion, which promotes monocyte infiltration and also polarization towards the immunosuppressive M2 phenotype [100].

Conversely, mutant p53 (mutp53) isoforms have aberrant proinflammatory functions, establishing chronic inflammatory circuits that drive tumor progression [101]. Mutp53 physically associates with HIF1α to stabilize hypoxia-inducible transcription programs, resulting in excessive extracellular matrix (ECM) deposition and fibroblast activation [102]. This remodeling creates a pro-inflammatory niche characterized by elevated TGF-β and SDF-1α signaling, which recruits cancer-associated fibroblasts (CAFs) and promotes EMT [103]. In the TME, cancer-associated fibroblasts (CAFs) play a vital role in controlling immune responses by gathering and organizing leukocytes [104]. Alterations in the p53 status of CAFs affect the tumor inflammatory milieu, with p53 mutations frequently detected in CAFs from highly inflamed cancers [105]. The mutp53-HIF1α axis further amplifies inflammatory signaling by inducing COX-2/mPGES-1 expression, establishing a positive feedback loop through prostaglandin E2 (PGE2) secretion that sustains NF-κB activation [106].

p53 modulates tumor-immune system crosstalk in the TME. p53 deletion or mutation affects the number of immune cells, such as bone marrow, tumor-associated macrophages, Tregs, and TH cells, which in turn exacerbate the pro-oncogenic transformations of inflammation [107]. p53 deletion leads to an increased circulation by regulating the WNT pathway [70]. The p53-deficient state in tumors increases the number of peripheral and intertumoral Treg cells, and the recruitment of Tregs leads to a decreased T-cell response [108]. Investigations into various cancers, including ovarian and lung cancers, revealed that p53 dysfunction prompts macrophages to overproduce pro-inflammatory cytokines, such as IL-1, IL-6, and IL-12, resulting in heightened local inflammation [109]. Immune cell dynamics are profoundly influenced by p53 status-dependent inflammatory cues. In wild-type contexts, p53-induced CXCL9/CXCL10 secretion facilitates Th1-polarized CD4+ T cell infiltration while suppressing Treg accumulation through miR-155-mediated regulation of FOXP3 [110]. This balance is disrupted in mutp53 tumors, where the elevated IL-23/IL-17 axis promotes neutrophil extracellular trap (NET) formation and neutrophil-mediated ECM degradation [111]. Simultaneously, mutp53 drives dendritic cell (DC) dysfunction through PD-L1 upregulation and MHC-II downregulation, creating an immunologically “cold” TME resistant to checkpoint blockade [112]. Tumor cells manipulate the alignment and orientation of the extracellular matrix (ECM) to promote tumor progression and interact with the hypoxic environment to amplify inflammatory signals [113]. In addition, p53 has been demonstrated to alter cancer cell secretion, which has implications for inflammatory response and cancer cell behavior. In non-small cell lung cancer, mutations in mutp53 that interact with HIF1-α cause an increase in ECM collagen VIIa1 and laminin-γ2 expression, which intensifies the release of inflammatory factors by activating the YAP/TAZ signaling pathway [114].

The inflammatory consequences of p53 dysfunction extend to metabolic reprogramming, with mutp53 enhancing glycolytic flux and lactate secretion via the stabilization of PKM2 tetramers [115]. This metabolic shift fuels tumor cell proliferation and establishes an acidic TME that impairs NK cell cytotoxicity and promotes M2 macrophage polarization [116]. The resulting inflammatory milieu is further exacerbated by mutp53-mediated suppression of senescence-associated secretory phenotype (SASP) resolution, creating a persistent inflammatory state that drives therapy resistance [117].

This duality in p53 function, that is, context-dependent suppression or promotion of inflammation, highlights its therapeutic tractability. Targeting mutp53-driven inflammatory circuits through inhibitors of NF-kB/HIF1α crosstalk or metabolic checkpoints represents a promising strategy to reverse immune exclusion in refractory malignancies [118] (Figure 4).

## 4. p53 and Cancer Metabolism

p53 exerts a direct or indirect regulatory influence on several metabolic processes, including glucose and glutamine catabolism [119], lipid and amino acid metabolism [120], and iron toxicity [121]. p53 senses metabolites and regulates their stability and activity. p53 regulates metabolic homeostasis in a direct or indirect manner by modulating key metabolic pathways to regulate the metabolic pathways that lead to metabolism [122]. It has been demonstrated that this process typically results in the direction of anabolism towards catabolism. When nutrients are scarce, p53 ensures the energy needed for cell survival and essential life functions, along with the materials necessary for biomolecule synthesis (Figure 5).

### 4.1. Glucose Metabolism

#### 4.1.1. Glycolysis

In cancer cells, glycolysis is often enhanced and it has been linked to the transformation of pyruvate into lactate [123]. Lactate is exported rather than directed into the mitochondria for ATP production, a process known as the Warburg effect [124]. p53 inhibits glycolysis in multiple steps [125,126]. Initially, it decreases glucose absorption by directly blocking the transcription of glucose transporter proteins such as GLUT1, GLUT4, and GLUT12 [127]. TP53 encodes the TP53-induced glycolysis and apoptosis regulator (TIGAR), which possesses fructose bisphosphates activity [128]. TIGAR subsequently lowers fructose 2,6-bisphosphate concentration, thereby decreasing the rate of glycolysis [129]. The rate of glycolysis is reduced when p53 facilitates the ubiquitination and inactivation of the glycolytic enzyme phosphoglycerate mutase (PGM) [130]. After glucose is transported into the cell, p53 inhibits a number of glycolytic enzymes, including hexokinase 1 (HK1), HK2, glucose-6-phosphate isomerase (GPI), PGM, and β-enolase [131,132,133]. This occurs via the repression or induction of certain microRNAs (miRNAs) [134]. Consequently, the activation of p53 restricts the flow of glucose to pyruvate, which is the end product of glycolysis.

#### 4.1.2. Gluconeogenesis

Gluconeogenesis is the process by which cells synthesize glucose. Part of this process is an inverse reaction, and p53 can inhibit or activate glycolysis [135]. Regulation of glucose levels may be environmentally and organizationally relevant. Studies have shown that p53 can induce genes encoding enzymes involved in gluconeogenesis such as glucose-6-phosphatase, phosphoenolpyruvate carboxykinase 2 (PCK2), glycerol kinase (GK), and aquaporin 3 (AQP3), which are involved in the process of gluconeogenesis [136,137]. p53 also activates pantothenate kinase 1 (PANK1), resulting in increased intracellular levels of coenzyme A (CoA) and gluconeogenesis [138]. However, p53 has also been reported to inhibit G6PC and PCK1 via SIRT6, thus inhibiting gluconeogenesis [139]. In cases of hepatocellular carcinoma linked to hepatitis B infection, p53 reduces the synthesis of glycogen synthase 2 (GYS2), which in turn lowers glycogen levels [140].

#### 4.1.3. Tricarboxylic Acid Cycle

The tricarboxylic acid (TCA) cycle, followed by oxidative phosphorylation, enables the complete decomposition of biomolecules in order to generate ATP. Typically, p53 decreases glycolysis, which hinders cancer development by enhancing pyruvate conversion. Pyruvate dehydrogenase (PDH) converts pyruvate to acetyl-CoA, which is essential for pyruvate to enter the TCA cycle [66]. Notably, when the glucose supply is limited, p53 promotes glutamine hydrolysis by upregulating the expression of glutaminase 2 (GLS2), which introduces glutamine into the TCA cycle [138]. The promotion of the TCA cycle also enhances mitochondrial respiration. As the TCA cycle and OXPHOS occur in the mitochondria, p53 limits glycolysis, transfers pyruvate to the TCA cycle, and facilitates OXPHOS by maintaining mitochondrial structural integrity and genomic stability [141]. p53 increases the transcription of various components within the respiratory chain complex, including cytochrome oxidase 2 (SCO2) and apoptosis-inducing factors (AIF) [142]. DPYSL4, a target of p53, associates with the mitochondrial super-complex to enhance OXPHOS [143]. p53 moves into the mitochondria and attaches to oligomycin-sensitive conferring protein (Oscp), aiding the formation of F_1_F_0_-ATP synthase [144]. Furthermore, p53 attaches to the NF-κB subunit RelA (also referred to as p65), preventing its movement to the mitochondria and eliminating the RelA-induced suppression of OXPHOS [145]. In other words, p53 inhibits tumorigenesis by reducing the cellular reliance on glycolysis and promoting oxidative phosphorylation.

#### 4.1.4. The Insulin–p53 Axis in Metabolic Dysfunction and Cancer Risk

Dysregulation of insulin signaling can trigger metabolic diseases such as diabetes, which is characterized by either insufficient insulin production or an impaired response to insulin [146]. These conditions are associated with several significant risk factors [147]. For example, hyperinsulinemia associated with metabolic diseases may increase the risk of cancer and mortality. Research has demonstrated that non-alcoholic fatty liver disease (NAFLD) can progress to fibrosis, cirrhosis, and hepatocellular carcinoma (HCC) [148]. This progression is often linked to persistent activation of the PI3K/AKT pathway, which subsequently inhibits p53 tumor suppressor activity [149]. Conversely, p53 regulates insulin secretion via several mechanisms. First, p53 plays a vital role in the regulation of glucose metabolism. Second, p53 influences pancreatic function and beta cell survival. Its activity is induced in beta cells of diabetic patients and in rodent models of type 2 diabetes, where it is crucial for beta cell proliferation and survival [150]. p53 contributes to insulin sensitivity regulation. Insulin resistance, a key indicator of pre-diabetes and type 2 diabetes, has been shown to be associated with oxidative stress and inflammation. Studies have shown that SREBP-1c, a key regulator of fatty acid synthesis, is downregulated by p53, highlighting its role as an essential repressor of fat production [151]. This indicates that significantly elevated p53 levels are closely associated with obesity-induced insulin resistance. Therefore, dysregulation of the insulin–p53 axis represents a key mechanistic link that explains the established association between metabolic disorders and cancer development, forming a dangerous feed-forward loop.

### 4.2. Lipid Metabolism

Lipids serve as membrane components, energy sources, and signaling messengers [152]. Cancer cells enhance lipid uptake and synthesis to fuel their rapid proliferation and increase fatty acid availability for membrane building and energy [153]. p53 counteracts these pro-tumorigenic processes by simultaneously targeting lipid breakdown and synthesis. First, p53 transcriptionally boosts the expression of three carnitine acyltransferases, carnitine octanoyltransferase (CROT), CPT1A, and CPT1C. These enzymes assist in fatty acid transport in response to nutrient levels [154]. p53 plays an important role in controlling cholesterol metabolism and the mevalonate (MVA) pathway, which is an important function of p53 [155,156]. p53 enhances the expression of ATP-binding cassette subfamily A member 1 (ABCA1) in hepatocellular carcinoma [157]. This blocks SREBP-2 maturation and inhibits MVA [158]. Conversely, mutp53 in cancers like breast cancer activates MVA via sterol regulatory element-binding protein (SREBP), accelerating tumor progression [159,160]. In addition, p53 is involved in lipid metabolism by regulating aromatase and SIRT1 [161]. In general, the p53–lipid axis is involved in tumor suppression and the balance of glycolytic and respiratory pathways. It is characterized by the strategic targeting of MVA, a regulatory hub with profound therapeutic implications. This distinguishes it from metabolic adaptations that occur during other cycles.

### 4.3. Amino Acid Metabolism

p53 promotes amino acid production during glucose deprivation [162]. Cancer cells have a significantly increased need for certain amino acids, especially glutamine and serine, at this time [163]. Following serine starvation, p53 helps cells survive by inducing a transient cell cycle block mediated by p21 [164]. This response redirects serine from nucleotide production to the glutathione (GSH) biosynthesis pathway, counteracting ROS and promoting cancer cell survival [165]. p53 has two effects on glutamine catabolism [166]. p53 influences glutamine catabolism by modulating GLS2 expression, which reduces the sensitivity of cells to ROS-related apoptosis in a manner reliant on p53 [167]. Alternatively, p53 enables cancer cells to adapt to glutamine deficiency by maintaining aspartate utilization to support the cells [168]. This is achieved by enhancing the expression of the aspartate/glutamate transporter protein SLC1A3, which is crucial for sustaining the electron transport chain and TCA cycle activity [169,170]. Metabolic flexibility, which prioritizes survival over proliferation during amino acid stress, is a key vulnerability that can be exploited in therapies targeting glutamine dependence, such as GLS inhibitors, in p53-deficient tumors.

### 4.4. Nucleotide Metabolism

Cancer cells boost nucleotide production rates in order to facilitate rapid growth [171]. p53 directly or indirectly limits nucleotide biogenesis to exert oncogenic effects by inhibiting dTTP and GMP synthesis [172]. p53 indirectly inhibits GTP production by inducing microRNA-34a (miRNA-34a), which disrupts the synthesis of inosine monophosphate dehydrogenase (IMPDH) enzyme. p53-mediated inhibition and its impact on nucleotide synthesis hinder mitosis in cancer cells [173]. In addition to its other functions, p53 promotes nucleotide production, aiding DNA repair and genomic stability by activating p53R2 (RRM2B) to increase ribonucleotide reductase activity when DNA is damaged [174]. This dual role of suppressing nucleotides for proliferation while enabling their synthesis for repair positions p53 status as a critical biomarker for predicting tumor response to antimetabolite therapies and DNA-damaging agents.

### 4.5. Iron Metabolism

Cancer cells frequently modify iron metabolism, leading to increased iron storage within the cells, which subsequently promotes cell proliferation and metastasis [175]. p53 plays a pivotal role in this intricate network, affecting iron uptake, storage, and utilization at the systemic and cellular levels. This is achieved by reducing the ability of cells to regulate iron levels [176]. p53 controls the transcription of various important iron regulators, such as hemomodulin (HAMP), iron–sulfur cluster assembly (ISCU), ferric oxide reductase (FDXR), and fraxin (FXN) [177,178]. p53 reduces iron absorption by inhibiting the iron transport proteins transferrin receptor 1 (TFR1) and Zrt-and-Irt-like protein 14 (ZIP14) at the post-transcriptional level [179]. p53 also activates haemitropia, also known as haemotropin antimicrobial peptide (HAMP), which sequesters iron in reticuloendothelial cells and macrophages [178]. This leads to decreased plasma iron uptake and storage by the macrophages, resulting in lower plasma iron levels [180]. These three actions, mediated by p53, restrict iron availability in cancer cells, thereby curbing iron-dependent cancer proliferation and metastasis. Importantly, dysregulation of p53-mediated iron control represents a potential vulnerability that can be exploited by novel therapeutic strategies targeting iron metabolism in specific cancer subtypes.

In general, the numerous metabolic roles of p53 appear to be linked to how cells manage and also endure changes in nutrient availability [180]. Cells have developed intricate systems to detect and react to fluctuations in nutrient availability [181]. When nutrients are scarce, p53 induction offers multiple advantages: it can conserve energy by halting the cell cycle and suppressing cell growth while encouraging catabolic processes. This provides a basis for new discoveries in precision medicine.

## 5. p53 in Cancer Stem Cells and Therapeutic Resistance

### 5.1. p53 and Cancer Stem Cell (CSC) Stemness Maintenance

Stem cells are distinct cell types with the potential to self-renew and differentiate into mature tissue cells [182]. CSCs are tumor cells with features similar to those of stem cells [183]. CSCs can self-renew and differentiate, producing a heterogeneous array of cells in tumors [184]. CSCs are potential triggers of tumor growth and are involved in the recurrence and spread of various cancers [185]. p53 usually promotes differentiation and hinders dedifferentiation, thereby impairing CSC stemness and blocking tumor development [186]. Initial research demonstrated that p53 could inhibit the induction of the cancer stem cell (CSC) phenotype by binding to regulatory sequences within genes that encode established CSC markers, including CD44 and CD133. This inhibition occurs through the repression of transcription facilitated by the recruitment of histone deacetylase HDAC1 [187]. The expression of CD44 may reduce p53 stability by promoting the HER2-dependent activation of MDM2, potentially resulting in resistance in CSCs [188]. CD133 is crucial for preserving stemness in CSCs, and p53 suppresses CD133 by directly attaching to its promoter and attracting HDAC1 [189]. CD133 deletion inhibits core stem cell factors, such as Oct4, Nanog, Sox2, and c-Myc, and promotes differentiation [190]. Additionally, p53 indirectly induces miRNAs to impair CSC stemness. For example, p53 induces miR-34a to functionally target the CSC marker CD44, thereby inhibiting prostate cancer regeneration and metastasis [191]. In pancreatic cancer, the p53–miRNA–200 axis maintains a differentiated state by inhibiting Sox2- and NFATC1-mediated dedifferentiation [192]. In contrast, p53 GOF mutations enhance cancer stem cell properties [193]. GOF mutations in p53 directly upregulate the transcription of ALDH1A1, CD44, and LGR5, thereby promoting the maintenance of colorectal cancer CSCs [194]. Mutant p53 inhibits miR-130b and miR-194, deregulates Zeb1 and Bmi1, and promotes EMT and stem cell phenotypes [195].

### 5.2. p53 Mutations and Resistance Mechanisms

Mutant p53 proteins may influence different proteins in specific signaling pathways, in addition to being crucial for the creation and maintenance of CSCs. This modulation endows cancer cells with new characteristics, including resistance to anticancer therapies [196]. First, mutant p53 facilitates the expression of intracellular transporter proteins, indirectly enhancing cellular resistance to foreign killing drugs [197]. For example, MDR1, an important protein in the ABC family, is inhibited by wild-type p53 in normal cells. When normal cells experience drug-induced DNA damage, p53 triggers apoptosis and programmed cell death [198]. This function is completely lost in mutant p53 cells. CSCs abundantly express ABC transporter proteins that export drugs out of cells, conferring resistance to chemotherapy [199]. As aforementioned, TP53 increases the expression of anti-apoptotic receptors and CSC markers, thus reducing the ability of cancer cells to interfere with external drugs [200]. Furthermore, GOF mutant p53 enhances the levels of the anti-apoptotic proteins Bcl-2 and Bcl-xL while suppressing the pro-apoptotic proteins Bax, Bad, and Bid [201,202]. Some mutp53 has been reported to inhibit caspase-9- and p63/73-dependent induction of Bax and Noxa, leading to the anti-apoptotic effects of mutp53 and insensitivity to radiotherapy and chemotherapy in mutp53-carrying cells [203]. Wild-type p53 reduces the expression of several CSC markers, including CD44, c-KIT, NANOG, and OCT4 [204]. Changes in p53 levels lead to a loss of inhibition of these CSC markers, causing CSC transformation and heightened resistance to radiation and chemotherapy [205]. Mutations in p53 R273H reduce procaspase-3 levels, making chemotherapeutic drugs, such as methotrexate and adriamycin, ineffective at triggering apoptosis [206]. In colon cancer, mutp53 is unable to attach to the PUMA promoter to initiate its transcription [207]. This helps tumor cells evade apoptosis and reduces their sensitivity to 5-fluorouracil. The DNA repair mechanism mediated by p53 is impaired in most somatic cancer cells [208]. The DNA repair process is enhanced by mutp53, which stimulates CSCs to express high levels of DNA repair genes [209]. This aids CSCs in fixing the DNA damage induced by chemotherapy drugs [209]. 5-fluorouracil enhances the expression of p53 in colorectal cancer [210]. However, unlike wtp53, mutp53 cannot repress LRPPRC expression after DNA damage [211]. This leads to increased MDR1 transcription and, consequently, chemoresistance [212]. The status of p53 directly influences the cellular fate in the microenvironment of cancer stem cells [213]. Wild-type p53 is a classic tumor suppressor that participates in the maintenance of cellular homeostasis by regulating the degradation of related proteins. However, when p53 is mutated, its function undergoes a fundamental transformation, shifting from tumor suppressor to tumor promoter [214]. In the regulatory network depicted in the figure, mutp53 occupies a central position, directly or indirectly regulating key molecules such as CD44, EZH2, and FOXH1, thereby profoundly influencing the quiescent state and stemness maintenance of the cancer stem cells [214]. These molecules not only contribute to the undifferentiated state of cancer stem cells but also suppress the expression of differentiation genes, ensuring that cancer stem cells can maintain their long-term stemness characteristics and evade elimination by conventional therapeutic modalities [215]. p53 mutations are closely linked to the treatment resistance mechanisms of cancer stem cells [216]. Chemotherapy, one of the primary treatment modalities for cancer, relies heavily on achieving effective concentrations within cancer cells [217]. However, cancer stem cells can actively expel chemotherapeutic drugs from the cell via mechanisms involving ABC transporters and efflux pumps, thereby forming a robust defensive barrier [218]. p53 mutations, particularly mutp53, play a pivotal role in this process. Studies have shown that mutp53 can activate the PI3K/AKT signaling pathway, further upregulating the expression of molecules such as WIP and YAP/TAZ, thereby enhancing the function of ABC transporters and efflux pumps and exacerbating the efflux of chemotherapy drugs [219]. Additionally, the metabolic activity of ALDH is activated in the context of p53 mutations, metabolizing chemotherapeutic drugs and reducing their cytotoxicity, further exacerbating treatment resistance [220].

In summary, p53 mutations are pivotal in shaping the biological behavior and treatment resistance mechanisms of cancer stem cells. By unraveling the intricate interplay between p53 mutations and the key characteristics of cancer stem cells, such as quiescence, stemness maintenance, and treatment resistance, we can establish new theoretical frameworks and experimental insights for designing innovative therapeutic strategies targeting cancer stem cells (Figure 6).

## 6. Diagnostic and Therapeutic Implications

### 6.1. Diagnostic Biomarker Potential

DNA sequencing is the most accurate method for the identification of TP53 mutations [221]. Prior to sequencing, mutation screening assays using gels such as SSCP or PCR-RFLP are typically employed [222]. The amplification of the TP53 fragment is restricted by an enzyme to a site expected to be created or disrupted by the presence of the mutation. The resulting gel profile indicates the presence of a mutation, and the region is sequenced by direct sequencing methods [223,224].

Sanger sequencing was once the primary technique used for identifying TP53 mutations in clinical environments [225]. This technology provides a relatively simple and readily available sequencing method; however, sequencing the entire gene for clinical testing is unnecessary and impractical [226]. Almost all clinically relevant germline mutations can be identified using region-specific targeted Sanger sequencing or complementary DNA sequencing [227]. However, Sanger sequencing is time-consuming, lacks sensitivity in detecting small subclones carrying TP53 mutations, and has a detection limit of 10–20% mutant alleles [228]. This limits their use in the treatment of myelodysplastic syndrome (MDS), acute myeloid leukemia (AML), and other malignancies [229]. Next-generation sequencing (NGS) has become the gold standard for detecting gene mutations, such as those in TP53, owing to the widespread use of parallel sequencing technology in clinical diagnostic laboratories [230]. NGS can identify various mutation types with high analytical and diagnostic sensitivities while maintaining specificity [231]. Targeted NGS is strongly correlated with Sanger sequencing and can detect low-frequency mutations that Sanger sequencing cannot detect [232]. However, its sensitivity threshold depends on several variables, including the hardware used, testing method, and analytical pipeline [233]. Fluorescence in situ hybridization (FISH) allows visualization of deletions in specific chromosomal regions at the single-cell level through the binding of fluorescently labeled DNA probes to target sequences [234]. In particular, FISH has high analytical sensitivity for 17p and monosomy 17 [235]. However, FISH is limited to the assaying of predefined targets, thus failing to provide a comprehensive genome-wide profile, and it also potentially overlooks complex karyotypes [236]. Overexpression and accumulation of p53 protein are widely used to detect p53 abnormalities in tumors [237]. Immunohistochemistry is a simple technique that can be applied to preserved tissues in a manner similar to routine histopathological evaluations [238]. It does not require prior knowledge or assumptions regarding specific mutational events [235,239]. Assessing MDM2 and p53 staining in tumors provides information on transcriptional activity and functional inactivation in the absence of mutations [240]. The rationale behind this experiment was that mutant p53 cannot transcriptionally activate MDM2. This leads to the loss of the negative feedback loop and accumulation of p53 protein. This suggests that TP53 mutations typically exhibit diffuse, strong, positive nuclear staining, whereas the wild-type exhibits only weak, scattered staining [241]. The expression of p53 and its correlation with TP53 mutation status were examined in 143 patients with acute myeloid leukemia (AML) [242]. When the threshold was 0.5%, the sensitivity was 0.83, and the specificity was close to 0.89; however, when the threshold was 15%, the sensitivity was 0.5, and the specificity was 0.94 [243]. However, this approach has some limitations. Positive IHC does not recognize null mutations, including nonsense, deletion, insertion, or splice-junction mutations [244]. These mutations account for 30% of somatic TP53 mutations [245]. Past studies have demonstrated that IHC is ineffective in detecting TP53 mutations when the frequency of null mutations is high, raising doubts about its utility in clinical settings [246]. Furthermore, positive p53 immunostaining in early lesions should be interpreted cautiously as it may reflect wild-type protein accumulation within the inflammatory microenvironment or be due to aging. The predictive capacity of TP53 mutation status for tumor response, patient prognosis, and treatment outcomes has been evaluated in various cancer types, including breast cancer [247]. Most studies indicated a significant association between TP53 mutations and poor prognosis [248]. Furthermore, positive p53 immunostaining in early lesions should be interpreted with caution as it may reflect the accumulation of wild-type proteins within the inflammatory microenvironment or be the result of aging. The predictive value of the TP53 mutation status for tumor response, patient prognosis, and treatment outcomes has been assessed in various cancer types, including breast cancer [247]. Andersson et al. reported that TP53 mutation status is a significant factor in patients receiving adjuvant CMF therapy [249]. Time to recurrence-free survival (RFS), breast cancer-corrected survival (BCCS), and overall survival (OS) are critical prognostic indicators [249]. However, these studies have not conclusively established whether TP53 serves as an independent prognostic factor. Additionally, comparisons of individual mutations are constrained by insufficient statistical power owing to the varying classifications of mutations [247]. Moreover, there is no clear consensus regarding the specific mutations that confer a worse prognosis [250].

The role of TP53 mutations as therapeutic predictors cannot be generalized, perhaps because there is no single answer. However, studies exploring the feasibility of using TP53 mutations as predictors have been conducted for various types of cancers [251,252]. Adjuvant systemic therapies, including radiotherapy and tamoxifen-based hormone therapy, have demonstrated reduced efficacy in patients with TP53 mutations and positive lymph nodes [253]. Preliminary findings further suggest a potential association between TP53 mutations and diminished responses to fluorouracil-, adriamycin-, and cyclophosphamide-based chemotherapy regimens [254].

### 6.2. Therapeutic Strategy

#### 6.2.1. Restoring Wild-Type Activity

Adenoviral delivery of wtp53 to cancer cells is a straightforward strategy for restoring p53 activity [255]. Gendicine, which uses this principle, was the first gene therapy product approved for the treatment of a wide range of cancers [256]. Gendicine, when used in conjunction with chemotherapy and radiation therapy, generally yields significantly better results than standard therapy alone [257]. CRISPR/Cas9-mediated genome editing offers a direct therapeutic approach for tumor cells harboring p53 mutants [203]. In prostate cancer cells, the wild-type TP53 genotype and phenotype were reconstructed by replacing the non-functional TP53 414delC mutation site with a fully operational sequence [258]. CRISPR/Cas9 has also been used in a p53 gene sensor system, which specializes in the efficient killing of p53-deficient cancer cells [209]. Zhan et al. developed a p53 sensor capable of specifically identifying WTp53 expression in cells [259]. By integrating this p53 sensor with a diphtheria toxin using the CRISPR/Cas9 system, tumor cells deficient in p53 can be selectively eliminated. However, this treatment strategy carries a significant risk of off-target genomic alterations and lacks robust clinical validation, which must be carefully considered in clinical applications [260,261].

Restoring the wild-type functionality of mutp53 is an efficient approach for decelerating tumor advancement [262]. Research has demonstrated that certain small-molecule compounds and peptides, including CP-31398, RITA, PEITC, NSC319726, Chetomin, ReACp53, and pCAP, can restore the wild-type conformation of mutant proteins by refolding or preventing abnormal folding in the first place. APR-246, also referred to as PRIMA-1, belongs to a class of active methylene quinoline compounds [263]. APR-246 reestablishes the normal conformation and anti-tumor transcriptional function of mutant p53 by forming a covalent bond with its DNA-binding domain [264]. Evidence suggests that COTI-2 reactivates mutp53, restoring its DNA-binding properties, inhibits cell growth, and induces apoptosis [265]. In contrast, CP-31398 promoted the production of active p53 in tumors by increasing the thermodynamic stability of newly synthesized wild-type proteins [266].

Targeting molecular chaperones that protect mutant p53 from destabilizing it has also emerged as a viable therapeutic approach [267]. Molecular chaperones such as Hsp70/Hsp90 and DNAJA1 protect mutant p53 from degradation by E3 ubiquitin ligases such as MDM2 and CHIP. This protects mutant p53 and provides oncogenic functions. The complex formed between mutant p53 and Hsp90 also protects mutant p53 from degradation. Drugs such as SAHA and Trigonelline A resveratrol can promote Hsp90-dependent depletion of mutant p53 [268,269]. Through its CAAX domain and mevalonate pathway, DNAJA1 competitively blocks CHIP-mediated degradation, thereby protecting mutant p53 [270]. DNAJA1 antagonists, such as statins and GY122, reverse this process [271].

#### 6.2.2. Degradation of Mutp53

Emerging evidence has suggested that certain mutp53 variants can form stable intracellular aggregates that may contribute to oncogenic gain-of-function activities and cancer progression [272,273]. Thus, facilitating the degradation of mutp53 could potentially yield anti-tumor effects. Certain compounds, including garcinia cambogia, capsaicin, MCB-613, and NSC59984 have been shown to induce degradation of mutp53 [274]. Statins induce the degradation of misfolded mutp53 by inhibiting its interaction with DNAJA1 and inducing CHIP-mediated mutp53 degradation [275]. Nevertheless, these compounds exhibited minimal impact on wtP53 and DNA contact mutants. Research has indicated that targeting the degradation of mutp53 has significant therapeutic potential; however, more clinical trials are needed to confirm its effectiveness [276].

#### 6.2.3. Anti-MDM2 Agents

MDM2 serves as a distinctive E3 ubiquitin ligase protein that plays a key role in the ubiquitination and degradation of the p53 gene [277]. Consequently, various drugs and compounds have been developed to reactivate p53 and induce tumor cell death. These include MDM2 antagonists, drugs that block MDM2 expression, inhibitors of MDM2 ubiquitin ligase activity, inhibitors of MDM2 interaction with p53, and inhibitors of E3 ubiquitination of p53 [278,279]. However, achieving specificity for small-molecule inhibitors that interact with the MDM2 protein is difficult. Although some MDM2 inhibitors have entered clinical trials, none are currently commercially available [34].

#### 6.2.4. Targeting p53-Immune Axis for Cancer Therapy

As mentioned earlier, p53 is crucial for controlling both innate and adaptive immune responses, wtp53 is a vital component of the immune response mediated by Toll-like receptors [280]. Additionally, wtp53 contributes to activation of the MHC-I antigen presentation pathway by upregulating TAP1 [281]. However, mutations in TP53 impair T-cell recruitment and function, thereby facilitating immune evasion [282]. Although mutant p53 (mutp53) has been reported to upregulate PD-L1 expression in some contexts, potentially modulating immune evasion, its overall role in tumor immunogenicity remains complex and context-dependent [108]. Notably, this contrasts with the established role of p53 in promoting anti-tumor immunity.

#### 6.2.5. Disrupting Protein Interactions

Various studies have demonstrated that disrupting protein interactions, a form of synergistic lethal therapy, can serve as a therapeutic approach to target mutp53, which exerts its gain-of-function effects through interactions with numerous proteins [283,284,285]. Therefore, interfering with such interactions is a viable strategy. RETRA is a small molecule that frees p73 from the mutp53-p73 complex, thus inhibiting tumor growth [286]. Prodigiosin interferes with the interaction between mutp53 and p73 and boosts p73 expression, leading to an anti-tumor effect [287]. In addition to small molecules, short peptides can disrupt interaction between mutp53 and p73. Di Agostino et al. in their study demonstrated that SIMPs disrupted the mutp53-p73 interaction, restored p73-mediated transcriptional activity and apoptotic function, and enhanced the sensitivity of mutp53-positive tumor cells to adriamycin and cisplatin [288]. These strategies are summarized in Figure 7.

### 6.3. p53-Independent Therapeutic Strategy

Because of the high frequency of TP53 mutations and the development of resistance to p53-targeting therapies, innovative approaches that target cancer cells regardless of their p53 status have emerged as promising alternatives. Conventional strategies that reactivate wild-type p53 or target mutant p53 face significant challenges, including intrinsic heterogeneity in tumor p53 status, compensatory pathway activation, and therapy-induced resistance mutations. These limitations have catalyzed the development of agents that are independent of p53 functionality. Ling et al. reported a small molecule called FL118 that suppresses tumor development in a p53-independent manner [289]. FL118 exerts its p53-independent anticancer effects by simultaneously inhibiting the expression of key pro-survival proteins (survivin, Mcl-1, XIAP, and cIAP2), while also increasing the expression of pro-apoptotic proteins such as Bax and Bim, ultimately leading to cancer cell death by inhibiting proliferation and inducing apoptosis. This mechanism overcomes the limitations of p53 dysfunction commonly observed in advanced cancers. Furthermore, a follow-up study demonstrated that FL118 activates the p53 pathway by promoting the ubiquitination and degradation of MdmX [290]. In colorectal cancer cells with normal p53 function, FL118 targets and degrades MdmX via the Mdm2-MdmX complex, thereby triggering p21-mediated cellular senescence. However, in the absence of p53 or in cases with p53 mutations, FL118 failed to initiate the senescence program. Instead, it more potently induces caspase-dependent apoptosis by inhibiting anti-apoptotic proteins such as survivin and Mcl-1.

Furthermore, some past studies have utilized p53-independent mechanisms to regulate the fate of cancer cells. Yerlikaya et al. discovered that the proteasome inhibitors bortezomib and MG-132 induce p53-independent apoptosis in diverse human cancer cell lines [291]. This process is associated with the p53-independent induction of the pro-apoptotic protein Noxa. Similarly, evidence has demonstrated that the BH3-mimetic drug obatoclax can induce both apoptosis and autophagy-dependent cell death in B cell non-Hodgkin’s lymphoma cells. This process is facilitated by both p53-dependent and -independent mechanisms [292].

## 7. Challenges and Future Directions

The intricate biology of p53 continues to redefine the landscape of cancer research, offering unparalleled opportunities for precise diagnoses and therapeutic innovation [16]. Mutant p53 proteins often acquire gain-of-function activities that promote oncogenic traits such as metastasis, therapeutic resistance, and immune evasion, underscoring the urgency of developing strategies to counteract these effects [118].

Clinically, the translation of p53-focused research has become pivotal [34]. Advances in precision diagnostics, including liquid biopsy technologies for non-invasive mutation detection, AI-driven bioinformatics for functional annotation, and multi-omics integration for patient stratification, have enabled earlier cancer identification and a more accurate prognosis [293]. These tools not only facilitate the identification of patients likely to benefit from p53-targeted therapies, but also provide insights into the mechanisms of resistance and relapse. Small molecules capable of restoring wild-type p53 activity (e.g., APR-246 and COTI-2) or destabilizing mutant p53 conformations (e.g., MDM2/MDMX inhibitors and stapled peptides) have shown promise in preclinical and early phase clinical trials [294,295,296]. Combination approaches that leverage p53 modulation with chemotherapy, radiotherapy, and immunotherapy (e.g., PD-1/PD-L1 blockade) aim to exploit synthetic lethal interactions and, therefore, enhance treatment efficacy [297,298].

Several challenges and opportunities shape the future of p53-based oncologies. Firstly, the heterogeneity of p53 mutations necessitates tailored therapeutic strategies [299]. Different types of mutations have different effects and occur at different frequencies in different cancer types. The TP53 mutations in different cancers and their effects are summarized in Table 1. New technologies such as whole-genome sequencing, single-cell RNA sequencing, and AI have made it possible to evaluate the genomic and transcriptional profiles of patients with tumors and enable precision medicine [300]. Additionally, precise and effective targeted p53 therapy has become a major issue in gene therapy, and p53-independent tumor treatment regimens are also being actively explored. Second, the dynamic interplay between p53 and the tumor immune microenvironment remains underexplored. Future studies should investigate how the p53 status influences immune cell infiltration, cytokine networks, and checkpoint expression, with the goal of optimizing immunotherapy combinations [6]. Third, integration of real-world evidence (RWE) from clinical trials and electronic health records is critical for validating p53 biomarkers and refining treatment algorithms [301].

## 8. Conclusions

p53 is a linchpin of cancer biology that bridges fundamental mechanisms with translational applications. This field is poised to overcome long-standing barriers and deliver transformative therapies by harnessing advances in diagnostics, drug discovery, and systems biology. The next decade may witness the first FDA-approved p53-targeted drugs alongside diagnostic platforms that enable the real-time monitoring of p53 functional status. Ultimately, a deeper understanding of the multifaceted roles of p53 will not only redefine cancer classification but also pave the way for precision oncology paradigms that maximize patient survival and quality of life.

## Figures and Tables

**Figure 1 biomolecules-15-01088-f001:**
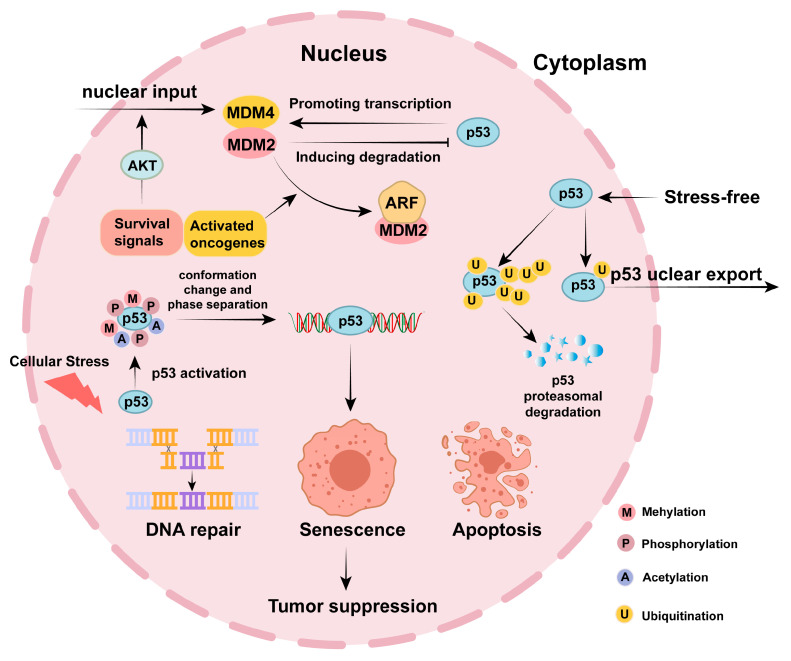
p53 expression control.

**Figure 2 biomolecules-15-01088-f002:**
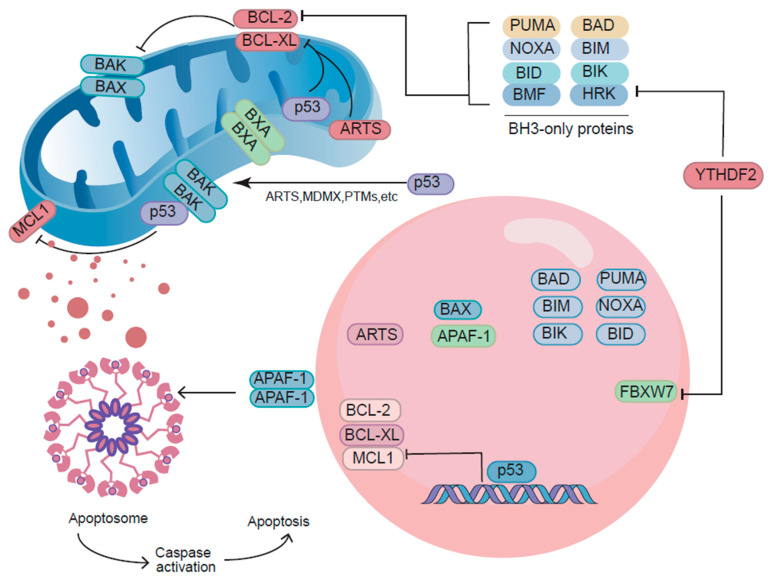
The mitochondrial apoptotic signaling network that relies on p53.

**Figure 3 biomolecules-15-01088-f003:**
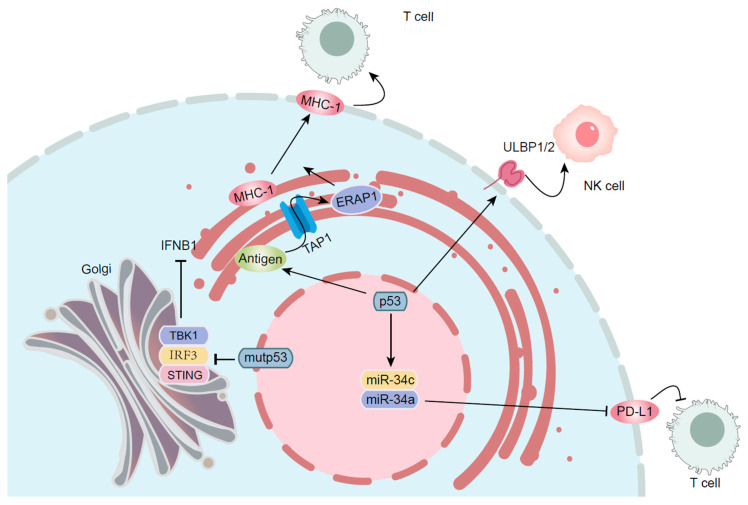
p53 regulates the immune system’s responses.

**Figure 4 biomolecules-15-01088-f004:**
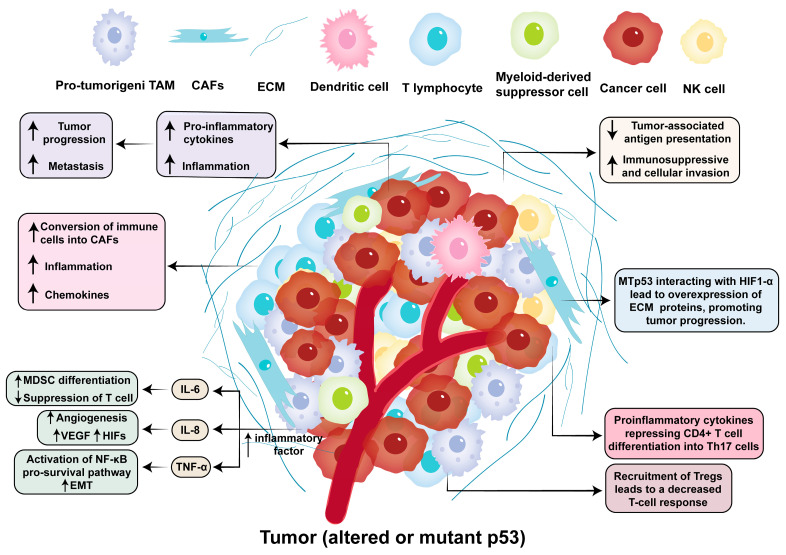
The malfunction of p53 influences its immune environment.

**Figure 5 biomolecules-15-01088-f005:**
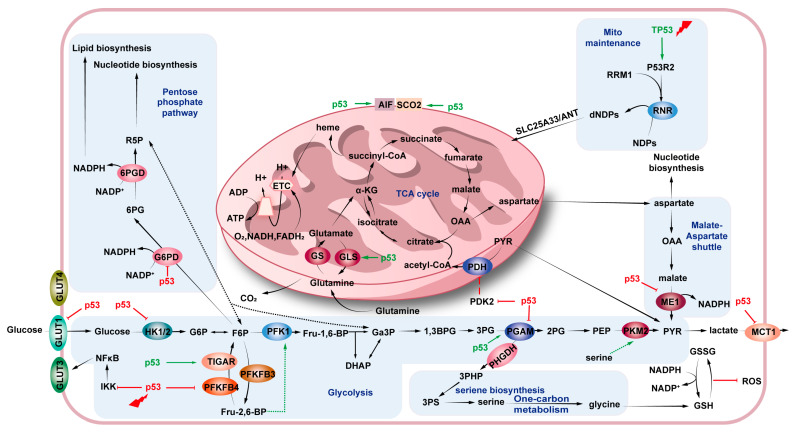
The regulation of glycolysis and mitochondrial functions by p53.

**Figure 6 biomolecules-15-01088-f006:**
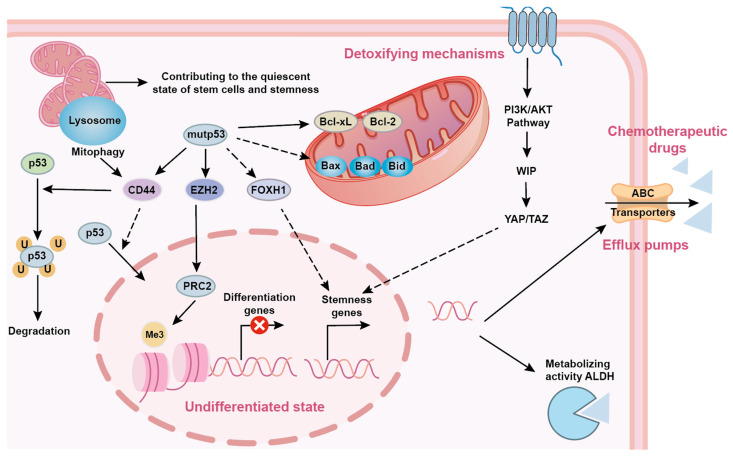
Interaction of stemness properties with mutp53.

**Figure 7 biomolecules-15-01088-f007:**
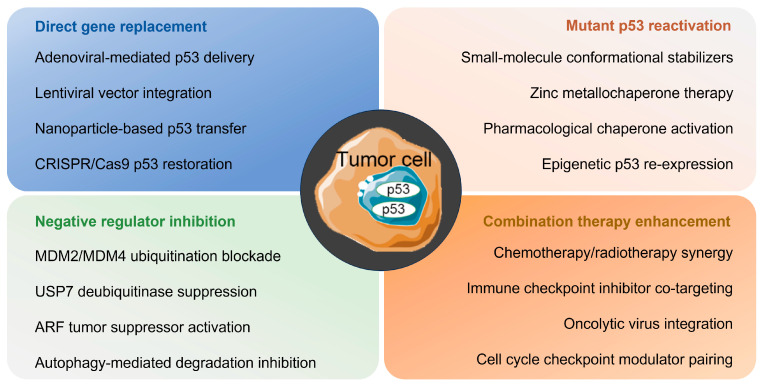
Anti-tumor treatments involving p53.

**Table 1 biomolecules-15-01088-t001:** Tp53 mutations in some common cancers and the effects they cause.

Cancer Type	p53 Mutation	Consequence	Reference
Colorectal cancer	R248Q	Hyperactivation of Jak2/Stat3 signaling to enhance tumor size, diversity, and aggressiveness.	[302]
	R273H/R273H	Increased secretion of inflammatory cytokines through sustained activation of NF-κB exacerbates chronic inflammation and invasion.	[303]
	R273H	Enhancement of invasiveness by promoting NF-κB activation or inhibiting the activation of ASK1/JNK by TNFα.	[304]
	R248W	Release of miR-1246-containing exosomes to promote the formation of uniquely reprogrammed macrophage populations.	[305]
	R273H/P309S	Enhancing tumor CSC marker-dependent chemoresistance and tumorigenesis (specifically ALDH1A1) through transcriptional modulation.	[306]
	R273H	Promote tumor CSCs marker expression to enhance tumor migration, invasion and chemotherapy resistance.	[307]
	R172H/R273H	Deregulates transcriptional repression of PDGFRβ by disrupting the p73/NF-Y complex, thereby promoting autonomous tumor cell invasion and metastasis.	[308]
Lung cancer	R175H/R175H/W	Indirect activation of the Axl promoter upregulates its expression to enhance tumor proliferation and migration.	[309]
	R175H/R248W/R282W/R273H	Enhancement of the REGγ-20S proteasome pathway in cancer cells, thereby affecting drug resistance and cell proliferation.	[310]
	R172H	Stabilization of DNA replication forks by increasing excitation of DNA replication starting points promotes proliferation of genomically abnormal cells.	[311]
	3KR	Regulation of cystine metabolism and iron-dead cells through the ability to inhibit SLC7A11 expression.	[312]
Breast cancer	R273H/R280K/L194F	Non-sequence-specific DNA binding or recruitment via other DNA-binding proteins to participate in the regulation of chromatin structure.	[313]
	R175H/R248Q/R248W/R249S/R273H	Promoting tumorigenesis and drug resistance by activating oncogenic transcription in an indirect manner.	[314]
	R175H/R280K/R273H	Enhanced survival and migration of cancer cells dependent on TXN upregulation and TXN/HMOX1 imbalance.	[315]
	R280K/R282W	Obstruction of KLF17 binding to target gene promoter repression leads to EMT repressor genes thereby enhancing invasion.	[316]
	R172H	Attenuates TBK1-dependent activation of IRF3, thereby promoting tolerance to cytoplasmic DNA and immune evasion.	[317]
Colon adenocarcinoma	R273H	Inhibition of the expression of the naturally occurring anti-inflammatory cytokine sIL-1Ra to promote tumor malignancy.	[318]
Oesophageal cancer	R273H/R175H	Binding to NRF2 directly promotes the accumulation of ROS in cancer cells and triggers death.	[319]
Head and neck squamous cell carcinoma	G245D/C238F/R175H	Promotes tumor invasion by inhibiting AMPK phosphorylation and deregulating FOXM1 transcriptional repression by FOXO3a.	[320]
Ovarian cancer	R135H	Upregulation of CCNG1 by inducing Notch3 expression promotes EMT, tumor metastasis and cisplatin resistance.	[321]

## Data Availability

No new data were created or analyzed in this study.

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
