# Peer review of "The Multifaceted Role of p53 in Cancer Molecular Biology: Insights for Precision Diagnosis and Therapeutic Breakthroughs"

_biomolecules, 2025, doi:10.3390/biom15081088_

Round 1
Reviewer 1 Report
Comments and Suggestions for Authors
With great curiosity, I checked out your manuscript. The subject—the several roles that p53 plays in tumor biology—is current and extremely pertinent. The range of p53's molecular functions, post-translational alterations, interactions with the tumor microenvironment (TME), immune system modulation, and consequences for cancer diagnostics and treatment is impressively covered in your review. Ferroptosis, metabolic rewiring, and stemness modulation are among the unique components that are highlighted in this well-referenced account of recent studies. Your efforts to synthesize a large and intricate corpus of literature are admirable.
However, several key issues must be addressed before the manuscript can be considered for publication:
1- The manuscript is thorough, although there are a few places where it lacks coherence and logical flow. For instance, several subsections (such as those on the metabolism of glucose, amino acids, lipids, and iron) create redundancy by repeating comparable chemical points. The sections should be more clearly organized around the main themes, with subheadings that rationally lead the reader from the traditional functions of p53 to its newly discovered involvement in precision medicine.
2-The English language throughout the manuscript requires substantial revision. Many sentences are overly complex or grammatically flawed, which hinders comprehension. For example:
- “Mutp53 upregulates PDL-1 expression and promotes CD8+ T-cell infiltration, thereby enhancing tumor immunogenicity.” — this statement is counterintuitive to current findings and needs clarification with references.
Please consider thorough language editing by a native English-speaking scientific editor.
3-Certain claims are presented as consensus but lack sufficient support. Statements such as:
“Mutp53 generates stable aggregates that build up within cells and significantly contribute to the progression of cancer.”
Should be backed by precise citations or qualified with phrases like “emerging evidence suggests...”.
Additionally, while therapeutic compounds like APR-246, Nutlin-3, and CRISPR-based strategies are described, the therapeutic limitations (e.g., off-target effects, trial failures, resistance) are underexplored. A balanced view is important
4-· Abbreviations such as MOMP, TAD, GOF, PTMs, and others should be defined upon first use.
5-Typographical errors are scattered throughout (e.g., "clothing factors" instead of "co-factors" on page 11).
6-check citation consistency: some bracketed references appear without numbers or proper formatting.
Comments on the Quality of English LanguageWith major revision, this manuscript can make a significant contribution to the field of cancer biology. I encourage the authors to reorganize the manuscript for clarity, revise the language thoroughly, reduce redundancy, and improve analytical depth.
Author Response
Review 1
With great curiosity, I checked out your manuscript. The subject—the several roles that p53 plays in tumor biology—is current and extremely pertinent. The range of p53's molecular functions, post-translational alterations, interactions with the tumor microenvironment (TME), immune system modulation, and consequences for cancer diagnostics and treatment is impressively covered in your review. Ferroptosis, metabolic rewiring, and stemness modulation are among the unique components that are highlighted in this well-referenced account of recent studies. Your efforts to synthesize a large and intricate corpus of literature are admirable.
Thank you very much for your time and effort reviewing our paper and your positive comments. We had followed your suggestions to improve the quality of our paper. We have superimposed our answers to your comments and suggestions to facilitate the revision. These are our answers to your inquiries:
However, several key issues must be addressed before the manuscript can be considered for publication:
1- The manuscript is thorough, although there are a few places where it lacks coherence and logical flow. For instance, several subsections (such as those on the metabolism of glucose, amino acids, lipids, and iron) create redundancy by repeating comparable chemical points. The sections should be more clearly organized around the main themes, with subheadings that rationally lead the reader from the traditional functions of p53 to its newly discovered involvement in precision medicine.
Answer:
We thank the reviewer for this constructive feedback. To address the concerns regarding coherence, logical flow, and redundancy within the metabolism sections (glucose, amino acids, lipids, iron), we have significantly reorganized the discussion. Specifically, we have reorganized the discussion to highlight the distinct roles of p53 in each metabolic pathway. This restructuring focuses each subsection more sharply on the unique regulatory mechanisms and consequences of p53 within that specific metabolic context, minimizing repetition of general chemical points.
Moreover, we have added subheadings to help readers read this section more clearly (e.g., 4.1.1 Glycolysis, 4.1.2 Gluconeogenesis, 4.1.3 Tricarboxylic acid cycle).
We have integrated precision medicine concepts more deliberately within the conclusions of key subsections, highlighting the traditional function of p53 and its relevance to precision medicine (revised manuscript pages 13, lines 494-496): This dual role of suppressing nucleotides for proliferation while enabling synthesis for repair positions p53 status as a critical biomarker for predicting tumor response to antimetabolite therapies and DNA-damaging agents.
We have deleted some redundant statements to ensure readability of this review.
2-The English language throughout the manuscript requires substantial revision. Many sentences are overly complex or grammatically flawed, which hinders comprehension. For example:
“Mutp53 upregulates PDL-1 expression and promotes CD8+ T-cell infiltration, thereby enhancing tumor immunogenicity.” — this statement is counterintuitive to current findings and needs clarification with references.
Please consider thorough language editing by a native English-speaking scientific editor.
Answer:
We sincerely thank the reviewer for their meticulous evaluation and valuable feedback on the manuscript's language quality. We acknowledge that certain passages contained grammatical complexities that hindered readability. To comprehensively address this:
- We apologize for the unclear statement regarding mutp53 and PD-L1. This sentence has been revised for accuracy and context:
- While mutant p53 (mutp53) has been reported to upregulate PD-L1 expression in some contexts [1],potentially modulating immune evasion, its overall role in tumor immunogenicity remains complex and context-dependent. Notably, this contrasts with wild-type p53's established role in promoting anti-tumor immunity.
- We have conducted a full manuscript scrutiny and every section was reviewed to eliminate grammatical flaws, awkward phrasing, and ensure subject-verb agreement.
- The manuscript has undergone rigorous editing by specify service.
3-Certain claims are presented as consensus but lack sufficient support. Statements such as:
“Mutp53 generates stable aggregates that build up within cells and significantly contribute to the progression of cancer.”
Should be backed by precise citations or qualified with phrases like “emerging evidence suggests...”.
Additionally, while therapeutic compounds like APR-246, Nutlin-3, and CRISPR-based strategies are described, the therapeutic limitations (e.g., off-target effects, trial failures, resistance) are underexplored. A balanced view is important
Answer:
Thank you sincerely for this constructive suggestion. As recommended, we have made the following improvements:
- The statement regarding mutp53 aggregates has been revised with appropriate qualification and supporting references:
- Emerging evidence suggests that certain mutp53 variants can form stable intracellular aggregates, which may contribute to oncogenic gain-of-function activities and cancer progression [2,3].
- We have systematically reviewed the manuscript to identify similar claims and ensured all significant assertions, especially concerning mechanisms or consequences of mutp53, are now appropriately qualified and robustly referenced.
- Thank you for your thoughtful suggestions. After careful consideration, we have revised the discussion in section 6.2 of the original text to ensure the accuracy of the arguments presented. For example, in Section 6.2, we added the following statement, but are not limited to them (revised manuscript pages 18, lines 694-696): However, this treatment strategy carries significant risks of off-target genomic alterations and lacks robust clinical validation, which must be carefully considered in clinical applications [4,5]. (revised manuscript pages 19, lines 735-737) However, achieving specificity for small molecule inhibitors that interact with the MDM2 protein is difficult. While some MDM2 inhibitors have entered clinical trials, none are currently on the market [6].
4- Abbreviations such as MOMP, TAD, GOF, PTMs, and others should be defined upon first use.
Answer:
We sincerely apologize for the oversight in defining key abbreviations and appreciate this critical correction. All requested abbreviations (MOMP, TAD, GOF, PTMs) and other non-standard terms have been explicitly defined upon first use in the revised manuscript:
Term |
First use example |
Section |
MOMP |
mitochondrial outer membrane permeabilization (MOMP) |
2.3 |
TAD |
terminal transactivation domain (TAD) |
2.1 |
GOF |
gain-of-function (GOF) |
1 |
PTMs |
post-translational modifications (PTMs) |
2.2 |
CBP |
CAMP-response element binding protein |
2.1 |
NAFLD |
Non-alcoholic fatty liver disease |
4.1.4 |
5-Typographical errors are scattered throughout (e.g., "clothing factors" instead of "co-factors" on page 11).
Answer:
We sincerely apologize for the typographical errors. And the term "clothing factors" has been corrected. We have conducted a full review of the manuscript using spell check and manual specify service to resolve similar errors throughout the text. We have thoroughly spell-checked the entire article to avoid such errors.
6-check citation consistency: some bracketed references appear without numbers or proper formatting.
Answer:
Thank you for noting citation inconsistencies. All in-text references now follow consistent numbered bracket formatting. We have verified that every citation contains properly formatted numbers and corresponds to the reference list.
Comments on the Quality of English Language
With major revision, this manuscript can make a significant contribution to the field of cancer biology. I encourage the authors to reorganize the manuscript for clarity, revise the language thoroughly, reduce redundancy, and improve analytical depth.
Answer:
This reviewer made many helpful comments and suggestions, we appreciate the constructive feedback and we have made the following improvements in terms of language:
- We have reorganized content for logical flow. For instance, we have added five subheadings under 6.2. Therapeutic Strategy to ensure clarity in the discussion.
- We have thoroughly reviewed the entire article and carefully corrected the logical flow and conciseness of the argument, spelling and formatting errors, and references and lists. Moreover, we engaged professional editing services for comprehensive language revision
- After re-examining the review, we have deleted the redundant content discussed in the review. For example, we have merged the repetitive parts of section 7. (Challenges and Future Directions) and section 8. (Conclusion) and summarized them, significantly reducing the length of these two sections.
We believe these revisions significantly enhance the manuscript's scientific value and readability.
Reference
- Blagih, J.; Zani, F.; Chakravarty, P.; Hennequart, M.; Pilley, S.; Hobor, S.; Hock, A.K.; Walton, J.B.; Morton, J.P.; Gronroos, E.; et al. Cancer-Specific Loss of p53 Leads to a Modulation of Myeloid and T Cell Responses. Cell Rep 2020, 30, 481-496 e486, doi:10.1016/j.celrep.2019.12.028.
- Ferretti, G.D.S.; Quarti, J.; Dos Santos, G.; Rangel, L.P.; Silva, J.L. Anticancer Therapeutic Strategies Targeting p53 Aggregation. Int J Mol Sci 2022, 23, 11023, doi:10.3390/ijms231911023.
- Kwan, K.; Castro-Sandoval, O.; Gaiddon, C.; Storr, T. Inhibition of p53 protein aggregation as a cancer treatment strategy. Curr Opin Chem Biol 2023, 72, 102230, doi:10.1016/j.cbpa.2022.102230.
- Mirgayazova, R.; Khadiullina, R.; Chasov, V.; Mingaleeva, R.; Miftakhova, R.; Rizvanov, A.; Bulatov, E. Therapeutic Editing of the TP53 Gene: Is CRISPR/Cas9 an Option? Genes 2020, 11, 704, doi:10.3390/genes11060704.
- Newman, A.; Starrs, L.; Burgio, G. Cas9 Cuts and Consequences; Detecting, Predicting, and Mitigating CRISPR/Cas9 On- and Off-Target Damage: Techniques for Detecting, Predicting, and Mitigating the On- and off-target Effects of Cas9 Editing. Bioessays 2020, 42, e2000047, doi:10.1002/bies.202000047.
- Wang, H.; Guo, M.; Wei, H.; Chen, Y. Targeting p53 pathways: mechanisms, structures, and advances in therapy. Signal Transduct Target Ther 2023, 8, 92, doi:10.1038/s41392-023-01347-1.

Reviewer 2 Report
Comments and Suggestions for Authors
This is a very long review article which needs many hours or even a day to read. One option to significantly shorten it is by using already published representative review articles to summary the current situation and then make succinct update.
This review article has issues. Some typical examples and specific comments are listed below. This should be seriously considered before publication in Biomolecules by re-reviewing the revised review manuscript.
- May try to improve the writing. For example, “functioning both as a tumor suppressor and, depending on the context, as an oncogenic promoter.” may be revised as “functioning as both a tumor suppressor and an oncogenic promoter, depending on the context.” to be easier for reading.
- Again, this is a very long review article. Whether these authors could shorten it by only review the update but not talking about p53 history researches/studies that have been reviewed many times. For example, for the “2.1. Transcriptional Regulation”, these authors provide a series of examples from various research and review articles to show p53 indeed involved in various transcriptional regulation of various gene expression, etc. However, this reviewer is thinking that this is already demonstrated and reviewed many times, and these authors should use one or two short paragraphs with or without figures to summarize the previous findings mainly using review articles. Then showing update stuff for this review article to let readers learn new things. With a final paragraph summary with authors’ opinion (This review noted the final paragraph in this subsection is indeed something related this, a good starting point).
- The “Activating sirtuin (SIRT1)/SLC7A11 signaling pathway is one of the classical signaling pathways in iron death, SIRT1 is a class III histone deacetylase, which acts through nicotinamide adenine dinucleotide-dependent deacetylation of its target proteins, and SIRT1 is involved in the regulation of antioxidant, mitochondrial function, and anti-apoptosis [46,47].” is a typical example of a bad leading sentence for that paragraph without leading readers to expect what these authors will show the audience for p53’s “Post-Translational Modifications”, which is the focus of “2.2.” subsection.
Why not put a take-home-message sentence like … … with p53 or in any other format that can let readers know what these authors will talk about in this paragraph?
- What is uclear (nuclear?). Please check typo.
- Another example: The leading sentence (description itself is OK) under the first paragraph of “3.1. Angiogenesis Regulation” cited Ref 80 (80. Hermans, D.; Rodriguez-Mogeda, C.; Kemps, H.; Bronckaers, A.; Helga, E.D.; Broux, B. Pathological angiogenesis: mecha- 1070nisms and therapeutic strategies. Angiogenesis 2023, 26, 349-362, doi:10.1007/s10456-023-09871-y.). However, after checking this ref80 showed below.
----------------Review
Angiogenesis. 2023 Aug;26(3):349-362.
doi: 10.1007/s10456-023-09871-y.Epub 2023 Mar 3.
Nectins and Nectin-like molecules drive vascular development and barrier function
Doryssa Hermans 1, Carla Rodriguez-Mogeda 2, Hannelore Kemps 3 4, Annelies Bronckaers 3, Helga E de Vries 2, Bieke Broux 5
Affiliations expand; PMID: 36867287; DOI: 10.1007/s10456-023-09871-y------------------
Additionally, these authors cited Refs 81, 82 and 93 which has nothing to do with Cancer. This could lead to misleading if not appropriately explained and compared in normal/other diseases versus cancer.
- This reviewer also suggests that each paragraph leading sentence may not use a reference that does not closely link to p53, since this artile is a p53-focused Review article. A better way is to give a sentence that cover the entire paragraph content that could be expected by readers. This reviewer noted that in some paragraphs, these authors did this.
- Another point is that in each section/subsection or long paragraph at the end of the section/subsection or long paragraph, these authors should have a short summary paragraph with their knowledge-based opinion (when can do so) for readers to have an overall understanding with key info.
- Be consistency for all things including format in the entire article. For example. If you bold figures, all figures should be bolded in the text. But this reviewer saw the (Figure 6) not bolded.
- For the “6.2. Therapeutic Strategy” section, while such as the mutant p53-reactivating compounds could be a good strategy to restore p53 WT function (e.g., the work reviewed in the Refs 274, 275), a novel strategy to overcome p53-based therapy challenges is to consider the development of p53 WT and Mutant-independent antitumor agents to kill cancer cells. These authors could not miss a brief review of such out-of-box strategy that has already in the place for example using at least on paragraph at the end of the “6.2. Therapeutic Strategy” section or using a new subsection (proffered).
Based on what this reviewer knows, the following examples of agents using 53-independent pathway to regulate cancer cell fate are provided.
FL118: Ling X et al. A Novel Small Molecule FL118 That Selectively Inhibits Survivin, Mcl-1, XIAP and cIAP2 in a p53-Independent Manner, Shows Superior Antitumor Activity. PLoS One. 2012). Futhermore, a follw-up study demonstrated that while FL118 induced a p53-dependent senescence in CRC cells, cells with p53 null or p53 mutant, FL118 even exhibited higher antitumor and cancer cell killing efficacy (Ling X et al. FL118 Induces p53-Dependent Senescence in Colorectal Cancer Cells by Promoting Degradation of MdmX. Cancer Res. 2014).
YM155: Majera D, Mistrik M. Effect of Sepatronium Bromide (YM-155) on DNA Double-Strand Breaks Repair in Cancer Cells. Int J Mol Sci. 2020;21(24).
Additionally, there are publications that use p53-independent mechanisms to control cancer cell fate for agents Like Obatoclax, Thapsigargin and Bortezomib, etc.
These auhtors are encouraged to make a PubMed search to review such jump-out-of-box strategy facing current p53 cancer therapeutics challenges using a new subsection. This will make the final submisection of “Challenges and Future Directions” be more complete.
- The “7. Challenges and Future Directions” section should be shortened.
- The “8. Conclusion” section should be significantly shortened. This section should be as short as possible.
- Many references have format issues without or without missing things. For example, Ref168 “168. Nakayama, A.; Yokoyama, M.; Nagano, H.; Hashimoto, N.; Yamagata, K.; Murata, K.; Tanaka, T. Mechanism of Mutant p53 Using Three-Dimensional Culture on Breast Cancer Malignant Phenotype via SREBP-Dependent Cholesterol Synthesis Pathway.”; “266. Peng, Y.W.; Bai, J.P.; Li, W.; Su, Z.D.; Cheng, X.Y. Advancements in -Based Anti-Tumor Gene Therapy Research. Molecules 2024, 29, doi: ARTN 5315 1589, 10.3390/molecules29225315.”. Likely many others.
Can be improved.
Author Response
Review 2
This is a very long review article which needs many hours or even a day to read. One option to significantly shorten it is by using already published representative review articles to summary the current situation and then make succinct update.
This review article has issues. Some typical examples and specific comments are listed below. This should be seriously considered before publication in Biomolecules by re-reviewing the revised review manuscript.
May try to improve the writing. For example, “functioning both as a tumor suppressor and, depending on the context, as an oncogenic promoter.” may be revised as “functioning as both a tumor suppressor and an oncogenic promoter, depending on the context.” to be easier for reading.
Answer:
We sincerely thank you for your meticulous reading and valuable time invested in evaluating our manuscript. We appreciate your keen eye for language, greatly helping improve the readability of our review. Regarding the specific example you mentioned, we have implemented your suggested revision verbatim (revised manuscript pages 1, lines 17). To ensure comprehensive language quality, we have conducted a systematic review of the entire manuscript to identify and correct similar syntactic complexities. Moreover, the paper has undergone rigorous language polishing by language services.
Again, this is a very long review article. Whether these authors could shorten it by only review the update but not talking about p53 history researches/studies that have been reviewed many times. For example, for the “2.1. Transcriptional Regulation”, these authors provide a series of examples from various research and review articles to show p53 indeed involved in various transcriptional regulation of various gene expression, etc. However, this reviewer is thinking that this is already demonstrated and reviewed many times, and these authors should use one or two short paragraphs with or without figures to summarize the previous findings mainly using review articles. Then showing update stuff for this review article to let readers learn new things. With a final paragraph summary with authors’ opinion (This review noted the final paragraph in this subsection is indeed something related this, a good starting point).
Answer:
We sincerely appreciate this constructive feedback regarding article length and focus. As you mentioned, current research on p53 transcriptional regulation is already quite in-depth. Section 2. is rather basic and tedious, but not sufficiently profound. It is indeed necessary to provide readers who want to grasp the latest research progress with a new perspective. Your suggestions are very helpful for our article. We have made comprehensive improvements in the following areas:
- We have reorganized the language, reduced redundant discourse, and greatly improved the readability of the article.
- Previous version (2.1. Transcriptional Regulation)
p53, as an important transcription factor, regulates gene transcription by reactively binding to DNA elements and recruiting clothing factors, and is one of the central mechanisms of cellular emergency response and tumor suppression [1,2]. p53 regulates gene transcription by combining with DNA sequences in the genome (called p53 response element, p53 response element, p53RE) [3]. The TP53, which encodes the p53 protein, functions to inhibit cancer formation [4]. To mediate tumor suppression, p53 binds to specific DNA response elements and induces the expression of genes involved in one or more of the following processes: cell cycle arrest, DNA repair, apoptosis, senescence, autophagy, iron death, or metabolism [5,6]. The tumor suppressor p53 functions as a central regulator of cellular stress responses through its multifaceted transcriptional regulatory network, acting as a critical determinant of genomic stability and tumor suppression[7,8]. As a sequence-specific transcription factor, p53 modulates gene expression programs by recognizing consensus DNA elements (p53 response elements, p53RE) and recruiting co-regulatory complexes to target promoters. This transcriptional axis directly mediates tumor suppression by inducing context-dependent expression of genes involved in cell cycle arrest (e.g., CDKN1A/p21), DNA repair (GADD45 family), apoptosis (BAX, PUMA), senescence (CDKN2A/p16INK4a), and metabolic reprogramming [9,10].
- Revised version (revised manuscript pages 2, lines 84-91)
Transcriptional regulation of p53 has been extensively studied in recent decades. p53, as an important transcription factor, suppresses tumors through sequence-specific DNA binding to p53 response elements (p53REs) [1-4]. This well-established mechanism involves the activation of genes that control cell cycle arrest (CDKN1A/p21), DNA repair (GADD45 family), apoptosis (BAX and PUMA), senescence (CDKN2A/p16INK4a), and metabolic reprogramming [9,10]. The tumor suppressor p53 functions as a central regulator of cellular stress responses through its multifaceted transcriptional regulatory network, and acts as a critical determinant of genomic stability and tumor suppression [7,8].
- We have updated our discussion so that the latest novelty research can be read. Throughout our review, we have added many novel studies. We have listed only the new references added in Section 2.:
Reference |
Title |
Published year |
[11] |
The landscape of human p53-regulated long non-coding RNAs reveals critical host gene co-regulation |
2023 |
[12] |
P53 long noncoding RNA regulatory network in cancer development |
2023 |
[13] |
Structural basis for DNA damage-induced phosphoregulation of MDM2 RING domain |
2020 |
[14] |
Deciphering the PTM codes of the tumor suppressor p53 |
2022 |
[15] |
Post-Translational Modifications of Proteins Orchestrate All Hallmarks of Cancer |
2025 |
- We strongly agree with your suggestion to add relevant summary paragraphs after each subsection, which greatly enhances the innovation and readability of this review. And we have added summary paragraphs at the end of each subsection to help readers better understand this review's content. For example, in Section 2.2., we have added a summary (revised manuscript pages 4, lines 165-173): As shown in Figure 1, As illustrated in Figure 1, these covalent modifications integrate diverse cellular signals to fine-tune p53 stability, transcriptional output, and non-genomic functions. Critically, the deregulation of PTM networks observed in over 50% of human cancers creates unique therapeutic vulnerabilities [14,15]. We posit that mapping patient-specific PTM signatures will enable the precise targeting of p53 dysregulation, as cancers with SIRT1 overexpression may benefit from acetylation-focused therapies. Realizing this precision medicine paradigm requires the development of modulators of PTM writers/erasers, and diagnostic tools to decode the p53 modification profiles of individual patients.
The “Activating sirtuin (SIRT1)/SLC7A11 signaling pathway is one of the classical signaling pathways in iron death, SIRT1 is a class III histone deacetylase, which acts through nicotinamide adenine dinucleotide-dependent deacetylation of its target proteins, and SIRT1 is involved in the regulation of antioxidant, mitochondrial function, and anti-apoptosis.” is a typical example of a bad leading sentence for that paragraph without leading readers to expect what these authors will show the audience for p53’s “Post-Translational Modifications”, which is the focus of “2.2.” subsection.
Answer:
We sincerely appreciate your insightful observation regarding the structural coherence of Section 2.2. We agree that the original opening lacked a clear conceptual framework linking to the subsection's focus on PTMs. To address this issue, we have reorganized the structure of the argument and added the thematic topic sentence at the beginning of this paragraph: Acetylation constitutes a key regulatory switch in p53 post-translational modification networks, dynamically controlling its functional outcomes through site-specific lysine modifications. Furthermore, we have written a thorough article that ensures every paragraph now begins with clear conceptual positioning.
Why not put a take-home-message sentence like … … with p53 or in any other format that can let readers know what these authors will talk about in this paragraph?
Answer:
Thank you for highlighting this. We have revised all topic sentences to explicitly state the paragraph’s p53-focused content.
What is uclear (nuclear?). Please check typo.
Answer:
We apologize for the error. This has been corrected throughout the manuscript.
Another example: The leading sentence (description itself is OK) under the first paragraph of “3.1. Angiogenesis Regulation” cited Ref 80 (80. Hermans, D.; Rodriguez-Mogeda, C.; Kemps, H.; Bronckaers, A.; Helga, E.D.; Broux, B. Pathological angiogenesis: mecha- 1070nisms and therapeutic strategies. Angiogenesis 2023, 26, 349-362, doi:10.1007/s10456-023-09871-y.). However, after checking this ref80 showed below.
----------------Review
Angiogenesis. 2023 Aug;26(3):349-362.
doi: 10.1007/s10456-023-09871-y.Epub 2023 Mar 3.
Nectins and Nectin-like molecules drive vascular development and barrier function
Doryssa Hermans 1, Carla Rodriguez-Mogeda 2, Hannelore Kemps 3 4, Annelies Bronckaers 3, Helga E de Vries 2, Bieke Broux 5
Affiliations expand; PMID: 36867287; DOI: 10.1007/s10456-023-09871-y------------------
Additionally, these authors cited Refs 81, 82 and 93 which has nothing to do with Cancer. This could lead to misleading if not appropriately explained and compared in normal/other diseases versus cancer.
Answer:
We sincerely appreciate your meticulous review and the time invested in evaluating our manuscript. We apologize for any lack of clarity regarding the cited references and fully understand your concern about potential misinterpretation. Reference 80 contains an incorrect citation and does not clearly indicate the influence of angiogenesis on tumor development. To ensure rigor and accuracy, we have replaced the reference (the title of the replacement reference is “Role of Angiogenesis and Its Biomarkers in Development of Targeted Tumor Therapies”). References 81, 82 and 93 were cited to establish fundamental mechanisms of p53 regulation observed in non-cancer models. Although these studies demonstrate the existence of conserved molecular pathways relevant to oncogenesis, citing these references would mislead readers. We have replaced the original references with more appropriate ones. And we have carefully reviewed and deleted the parts of this passage that are unrelated to cancer in order to avoid conveying information that may be misleading. We thank you for emphasizing this crucial aspect of scholarly rigor. Your insight has significantly strengthened the review's precision and translational relevance to cancer biology.
This reviewer also suggests that each paragraph leading sentence may not use a reference that does not closely link to p53, since this artile is a p53-focused Review article. A better way is to give a sentence that cover the entire paragraph content that could be expected by readers. This reviewer noted that in some paragraphs, these authors did this.
Another point is that in each section/subsection or long paragraph at the end of the section/subsection or long paragraph, these authors should have a short summary paragraph with their knowledge-based opinion (when can do so) for readers to have an overall understanding with key info.
Answer:
Thank you for your valuable suggestions regarding paragraph structure and section summaries. We agree that enhancing thematic focus on p53 in topic sentences and providing concluding insights will improve readability and scientific rigor. We have now implemented these improvements throughout the manuscript.
Be consistency for all things including format in the entire article. For example. If you bold figures, all figures should be bolded in the text. But this reviewer saw the (Figure 6) not bolded.
Answer:
We sincerely appreciate your meticulous attention to formatting consistency throughout the manuscript and thank you for identifying the oversight regarding Figure 6. We have corrected this specific instance by applying bold formatting (page 15) to ensure uniformity with all other figure references in the text. Furthermore, we have conducted a comprehensive review of the entire document to verify consistent application of all formatting conventions including figure and table labeling, heading hierarchies, citation styles, and abbreviation definitions—and implemented necessary adjustments to eliminate any remaining inconsistencies, thereby enhancing the professional presentation and reader accessibility of the manuscript.
This revision ensures complete adherence to the journal's style guidelines while maintaining the scholarly integrity of the content. We are grateful for your valuable feedback, which has significantly improved the article quality.
For the “6.2. Therapeutic Strategy” section, while such as the mutant p53-reactivating compounds could be a good strategy to restore p53 WT function (e.g., the work reviewed in the Refs 274, 275), a novel strategy to overcome p53-based therapy challenges is to consider the development of p53 WT and Mutant-independent antitumor agents to kill cancer cells. These authors could not miss a brief review of such out-of-box strategy that has already in the place for example using at least on paragraph at the end of the “6.2. Therapeutic Strategy” section or using a new subsection (proffered).
Based on what this reviewer knows, the following examples of agents using 53-independent pathway to regulate cancer cell fate are provided.
FL118: Ling X et al. A Novel Small Molecule FL118 That Selectively Inhibits Survivin, Mcl-1, XIAP and cIAP2 in a p53-Independent Manner, Shows Superior Antitumor Activity. PLoS One. 2012).
Futhermore, a follw-up study demonstrated that while FL118 induced a p53-dependent senescence in CRC cells, cells with p53 null or p53 mutant, FL118 even exhibited higher antitumor and cancer cell killing efficacy (Ling X et al. FL118 Induces p53-Dependent Senescence in Colorectal Cancer Cells by Promoting Degradation of MdmX. Cancer Res. 2014).
YM155: Majera D, Mistrik M. Effect of Sepatronium Bromide (YM-155) on DNA Double-Strand Breaks Repair in Cancer Cells. Int J Mol Sci. 2020;21(24).
Additionally, there are publications that use p53-independent mechanisms to control cancer cell fate for agents Like Obatoclax, Thapsigargin and Bortezomib, etc.
These auhtors are encouraged to make a PubMed search to review such jump-out-of-box strategy facing current p53 cancer therapeutics challenges using a new subsection. This will make the final submisection of “Challenges and Future Directions” be more complete.
Answer:
Thank you for your insightful feedback and constructive suggestions regarding the therapeutic strategies targeting p53. We sincerely appreciate your thorough evaluation and apologize for not initially emphasizing the critical role of p53-independent approaches in overcoming therapy resistance. We fully agree with your perspective that exploring beyond p53-targeting agents is essential for comprehensive cancer therapeutics. To address this valuable point, we have now established a dedicated new subsection entitled "6.3. p53-Independent Therapeutic Strategy" under Section 6 (Diagnostic and Therapeutic Implications). We believe this addition significantly enhances the depth and translational relevance of our discussion on future therapeutic directions.
6.3. p53-Independent Therapeutic Strategy
Because of the high frequency of TP53 mutations and the development of resistance to p53-targeting therapies, innovative approaches that target cancer cells regardless of p53 status have emerged as promising alternatives. Conventional strategies that reactivate wild-type p53 or target mutant p53 face significant challenges, including intrinsic heterogeneity in tumor p53 status, compensatory pathway activation, and therapy-induced resistance mutations. These limitations have catalyzed the development of agents that are independent of p53 functionality.Ling et al. have reported a small molecule called FL118 that suppress tumor development via a p53-independent manner [16]. FL118 exerts its p53-independent anticancer effects by simultaneously inhibiting the expression of key pro-survival proteins (survivin, Mcl-1, XIAP, and cIAP2), while also increasing pro-apoptotic proteins like Bax and Bim, ultimately leading to cancer cell death through inhibited proliferation and induced apoptosis. This mechanism allows it to overcome the limitations of p53 dysfunction commonly found in advanced cancers. Futhermore, a follw-up study demonstrated that FL118 activates the p53 pathway by promoting the ubiquitination and degradation of the MdmX protein [17]. In colorectal cancer cells with normal p53 function, FL118 targets and degrades MdmX via the Mdm2-MdmX complex, thereby triggering p21-mediated cellular senescence. However, in the absence of p53 or in cases of p53 mutations, FL118 fails to initiate the senescence program. Instead, it more potently induces caspase-dependent apoptosis by inhibiting anti-apoptotic proteins, such as survivin and Mcl-1.
Furthermore, there are publications that utilize p53-independent mechanisms to regulate cancer cell fate for agents. Yerlikaya et al. have discovered that proteasome inhibitor bortezomib and MG-132 could induce p53-independent apoptosis in diverse human cancer cell lines [18]. And this process is associated with the p53-independent induction of the pro-apoptotic protein Noxa. Similarly, evidence has demonstrated that the BH3-mimetic drug Obatoclax has the capacity to induce both apoptosis and autophagy-dependent cell death in B-cell non-Hodgkin's lymphoma cells. The process is facilitated by both p53-dependent and -independent mechanisms [19].
The “7. Challenges and Future Directions” section should be shortened.
The “8. Conclusion” section should be significantly shortened. This section should be as short as possible.
Answer:
We sincerely thank you for your valuable guidance on enhancing the conciseness of the concluding sections of manuscript. We have implemented a thorough restructuring of Sections 7 ("Challenges and Future Directions") and 8 ("Conclusion") to address your concerns
Many references have format issues without or without missing things. For example, Ref168 “168. Nakayama, A.; Yokoyama, M.; Nagano, H.; Hashimoto, N.; Yamagata, K.; Murata, K.; Tanaka, T. Mechanism of Mutant p53 Using Three-Dimensional Culture on Breast Cancer Malignant Phenotype via SREBP-Dependent Cholesterol Synthesis Pathway.”; “266. Peng, Y.W.; Bai, J.P.; Li, W.; Su, Z.D.; Cheng, X.Y. Advancements in -Based Anti-Tumor Gene Therapy Research. Molecules 2024, 29, doi: ARTN 5315 1589, 10.3390/molecules29225315.”. Likely many others.
Answer:
We sincerely thank the reviewer for their careful reading of our manuscript and for identifying the formatting issues in the references. We acknowledge these errors and apologize for any oversight. We have now thoroughly checked all references and corrected these formatting inconsistencies, including ensuring complete journal information and proper DOI/accession number presentation. We appreciate the reviewer's diligence in helping us improve the accuracy and presentation of the reference list.
Reference
- Sammons, M.A.; Nguyen, T.T.; McDade, S.S.; Fischer, M. Tumor suppressor p53: from engaging DNA to target gene regulation. Nucleic Acids Res 2020, 48, 8848-8869, doi:10.1093/nar/gkaa666.
- Fischer, M. Gene regulation by the tumor suppressor p53 - The omics era. Biochim Biophys Acta Rev Cancer 2024, 1879, 189111, doi:10.1016/j.bbcan.2024.189111.
- Kearns, S.; Lurz, R.; Orlova, E.V.; Okorokov, A.L. Two p53 tetramers bind one consensus DNA response element. Nucleic Acids Res 2016, 44, 6185-6199, doi:10.1093/nar/gkw215.
- Monti, P.; Menichini, P.; Speciale, A.; Cutrona, G.; Fais, F.; Taiana, E.; Neri, A.; Bomben, R.; Gentile, M.; Gattei, V.; et al. Heterogeneity of TP53 Mutations and P53 Protein Residual Function in Cancer: Does It Matter? Front Oncol 2020, 10, 593383, doi:10.3389/fonc.2020.593383.
- Janic, A.; Valente, L.J.; Wakefield, M.J.; Di Stefano, L.; Milla, L.; Wilcox, S.; Yang, H.; Tai, L.; Vandenberg, C.J.; Kueh, A.J.; et al. DNA repair processes are critical mediators of p53-dependent tumor suppression. Nat Med 2018, 24, 947-953, doi:10.1038/s41591-018-0043-5.
- Li, W.-f.; Herkilini, A.; Tang, Y.; Huang, P.; Song, G.-b.; Miyagishi, M.; Kasim, V.; Wu, S.-r. The transcription factor PBX3 promotes tumor cell growth through transcriptional suppression of the tumor suppressor p53. Acta Pharmacologica Sinica 2021, 42, 1888-1899, doi:10.1038/s41401-020-00599-9.
- Chen, S.; Thorne, R.F.; Zhang, X.D.; Wu, M.; Liu, L. Non-coding RNAs, guardians of the p53 galaxy. Semin Cancer Biol 2021, 75, 72-83, doi:10.1016/j.semcancer.2020.09.002.
- Boutelle, A.M.; Attardi, L.D. p53 and Tumor Suppression: It Takes a Network. Trends Cell Biol 2021, 31, 298-310, doi:10.1016/j.tcb.2020.12.011.
- Coronel, L.; Riege, K.; Schwab, K.; Forste, S.; Hackes, D.; Semerau, L.; Bernhart, S.H.; Siebert, R.; Hoffmann, S.; Fischer, M. Transcription factor RFX7 governs a tumor suppressor network in response to p53 and stress. Nucleic Acids Res 2021, 49, 7437-7456, doi:10.1093/nar/gkab575.
- Liu, Y.; Su, Z.; Tavana, O.; Gu, W. Understanding the complexity of p53 in a new era of tumor suppression. Cancer Cell 2024, 42, 946-967, doi:10.1016/j.ccell.2024.04.009.
- Wang, H.; Guo, M.; Wei, H.; Chen, Y. Targeting p53 pathways: mechanisms, structures, and advances in therapy. Signal Transduct Target Ther 2023, 8, 92, doi:10.1038/s41392-023-01347-1.
- Fischer, M.; Riege, K.; Hoffmann, S. The landscape of human p53-regulated long non-coding RNAs reveals critical host gene co-regulation. Mol Oncol 2023, 17, 1263-1279, doi:10.1002/1878-0261.13405.
- Magnussen, H.M.; Ahmed, S.F.; Sibbet, G.J.; Hristova, V.A.; Nomura, K.; Hock, A.K.; Archibald, L.J.; Jamieson, A.G.; Fushman, D.; Vousden, K.H.; et al. Structural basis for DNA damage-induced phosphoregulation of MDM2 RING domain. Nat Commun 2020, 11, 2094, doi:10.1038/s41467-020-15783-y.
- Wen, J.; Wang, D. Deciphering the PTM codes of the tumor suppressor p53. J Mol Cell Biol 2022, 13, 774-785, doi:10.1093/jmcb/mjab047.
- Bruno, P.S.; Arshad, A.; Gogu, M.-R.; Waterman, N.; Flack, R.; Dunn, K.; Darie, C.C.; Neagu, A.-N. Post-Translational Modifications of Proteins Orchestrate All Hallmarks of Cancer. Life 2025, 15, 126, doi:10.3390/life15010126.
- Ling, X.; Cao, S.; Cheng, Q.; Keefe, J.T.; Rustum, Y.M.; Li, F. A novel small molecule FL118 that selectively inhibits survivin, Mcl-1, XIAP and cIAP2 in a p53-independent manner, shows superior antitumor activity. PloS one 2012, 7, e45571, doi:10.1371/journal.pone.0045571.
- Ling, X.; Xu, C.; Fan, C.; Zhong, K.; Li, F.; Wang, X. FL118 induces p53-dependent senescence in colorectal cancer cells by promoting degradation of MdmX. Cancer Res 2014, 74, 7487-7497, doi:10.1158/0008-5472.CAN-14-0683.
- Yerlikaya, A.; Okur, E.; Ulukaya, E. The p53-independent induction of apoptosis in breast cancer cells in response to proteasome inhibitor bortezomib. Tumour Biol 2012, 33, 1385-1392, doi:10.1007/s13277-012-0386-3.
- Czuczman, M.; Gruber, E.; Hosking, P.; Olejniczak, S.H.; Khubchandani, S.; Hernandez-Ilizaliturri, F.J. The BH3-Mimetic Obatoclax (GX15-070) Posses a Dual-Mechanism of Action and Induces Both Apoptosis and Autophagy-Dependent Cell Death of B Cell Non-Hodgkin’s Lymphoma (B-NHL) Cells. Blood 2008, 112, 605-605, doi:10.1182/blood.V112.11.605.605.

Reviewer 3 Report
Comments and Suggestions for Authors
This review article discusses p53 role in cancer molecular biology for diagnosis and therapy.
Main concerns and comments:
- Please discuss pathways and/or mechanisms which can restore p53 function of p53 mutants especially loss of function
- Please discuss the relationship between insulin and p53 in metabolic diseases which have potentials impact in cancer development
- Please discuss p53 and gut microbiome in cancer development
Author Response
Review 3
This review article discusses p53 role in cancer molecular biology for diagnosis and therapy.
Main concerns and comments:
1. Please discuss pathways and/or mechanisms which can restore p53 function of p53 mutants especially loss of function
Answer:
We sincerely thank the reviewer for raising this valuable point regarding pathways to restore wild-type function in p53 mutants, particularly those exhibiting loss-of-function. This insightful comment is highly valuable as it elevates the manuscript's depth by specifically addressing a critical therapeutic strategy – functionally rescuing mutant p53 rather than solely eliminating it. Enhancing the discussion on this topic significantly improves the reader's understanding of potential clinical approaches for tumors harboring these prevalent mutations. Although there is some discussion of restoring p53 function in p53 mutants in Section 6.2 of the original text, the discussion is not clear or specific enough. After reviewing the literature, we have created a separate subheading (6.2.1. Restoring wild-type activity) to discuss this topic:
6.2.1. Restoring wild-type activity
Restoring the wild-type functionality of mutp53 represents an efficient approach to decelerate tumor advancement [1]. Research has demonstrated that certain small molecule compounds and peptides, including CP-31398, RITA, PEITC, NSC319726, Chetomin, ReACp53, and pCAP, can restore the wild-type conformation of mutant proteins by helping them refold or preventing abnormal folding in the first place. APR-246, which is also referred to as PRIMA-1, belongs to a class of active methylene quinoline compounds [2]. APR-246 reestablishes the normal conformation and antitumor transcriptional function of mutant p53 by forming a covalent bond with its DNA-binding domain [3]. Evidence suggests that COTI-2 reactivates mutp53, restoring its DNA-binding properties and inhibiting cell growth and inducing apoptosis [4]. In contrast, CP-31398 promotes the production of active p53 in tumors by increasing the thermodynamic stability of newly synthesized wild-type proteins [5].
Adenoviral delivery of wtp53 to cancer cells is a straightforward strategy for restoring p53 activity [6]. Gendicine, which utilizes this principle, was the first gene therapy product approved to treat a wide range of cancers [7]. Gendicine, when used in conjunction with chemotherapy and radiation therapy, generally yields significantly better results than standard therapy alone [8]. CRISPR/Cas9-mediated genome editing offers a direct therapeutic approach for tumor cells harboring p53 mutants [9]. In prostate cancer cells, the wild-type TP53 genotype and phenotype were reconstructed by replacing the non-functional TP53 414delC mutation site with a fully operational sequence [10].
Targeting the molecular chaperones that protect mutant p53 in order to destabilize it has also emerged as a viable therapeutic approach [11]. Molecular chaperones such as Hsp70/Hsp90 and DNAJA1 protect mutant p53 from degradation by E3 ubiquitin ligases like MDM2 and CHIP. This protects mutant p53 and gives it oncogenic functionality. The complex formed between mutant p53 and Hsp90 also protects mutant p53 from degradation. Drugs such as SAHA, Trigonelline A and resveratrol can promote Hsp90-dependent mutant p53 depletion [12]. Through its CAAX domain and the mevalonate pathway, DNAJA1 competitively blocks CHIP-mediated degradation, thereby protecting mutant p53 [13]. DNAJA1 antagonists, such as statins and GY122, can reverse this process [14].
2. Please discuss the relationship between insulin and p53 in metabolic diseases which have potentials impact in cancer development
Answer:
We sincerely thank the reviewer for their insightful comment highlighting the relationship between insulin signaling and p53 in metabolic diseases and its potential impact on cancer development. We greatly appreciate the reviewer's expertise in recognizing this critical connection, which underscores the fundamental importance of insulin pathways in glucose metabolism and how dysregulation in glucose metabolism-related metabolic diseases contributes to tumorigenesis – a key aspect we had not sufficiently emphasized in our initial draft. This valuable suggestion significantly enriches the depth and comprehensiveness of our review. In direct response to this point, we have carefully integrated this crucial discussion within Section 4.1. (Glucose Metabolism) by adding a dedicated new subsection, "4.1.4. The Insulin-p53 Axis in Metabolic Dysfunction and Cancer Risk". We believe this substantial addition significantly strengthens the review's scope and provides a more complete picture of the metabolic links to cancer, particularly concerning p53.
4.1.4 The Insulin-p53 Axis in Metabolic Dysfunction and Cancer Risk
Dysregulation of insulin signaling can trigger metabolic diseases like diabetes, characterized by either insufficient insulin production or impaired response to insulin [15]. These conditions involve significant risk factors [16]. For instance, hyperinsulinemia associated with metabolic diseases may increase cancer risk and mortality. Research has demonstrated that non-alcoholic fatty liver disease (NAFLD) can progress to fibrosis, cirrhosis, and hepatocellular carcinoma (HCC) [17]. This progression is often linked to the persistent activation of the PI3K/AKT pathway, which subsequently inhibits p53 tumor suppressor activity [18]. Conversely, p53 also regulates insulin secretion through several mechanisms. Firstly, as mentioned, p53 plays a vital role in regulating glucose metabolism. Secondly, p53 influences pancreatic function and beta-cell survival. Its activity is induced in the beta cells of diabetic patients and rodent models of type 2 diabetes, where it proves crucial for beta-cell proliferation and survival [19]. Finally, p53 itself contributes to regulating insulin sensitivity. Insulin resistance, a key indicator of prediabetes and type 2 diabetes, underlies oxidative stress and inflammation. Studies show that SREBP-1c, a key regulator of fatty acid synthesis, is downregulated by p53, highlighting its role as an essential repressor of fat production [20]. This indicates that significantly elevated p53 levels are closely associated with obesity-induced insulin resistance. Therefore, dysregulation of the insulin-p53 axis represents a key mechanistic link explaining the established association between metabolic disorders and cancer development, forming a dangerous feed-forward loop.
3. Please discuss p53 and gut microbiome in cancer development
Answer:
We sincerely thank the reviewer for raising this important question regarding the relationship between the gut microbiome and p53 in cancer development. This is indeed a highly significant and rapidly evolving area of research that offers profound insights into the microenvironmental influences on tumor suppression. We acknowledge that our current manuscript does not contain a dedicated section comprehensively covering this specific interaction, for which we apologize. The intricate bidirectional crosstalk between host p53 status and gut microbial composition, representing a novel and fascinating layer of complexity in cancer biology, particularly relevant for gastrointestinal malignancies. We appreciate the reviewer highlighting this cutting-edge topic, which further enriches the scope of our review and underscores the multifaceted nature of p53 regulation in oncogenesis.
In recent years, the interplay between gut microbiota and the development of intestinal neoplasms has emerged as a pivotal research focus. A substantial body of research has demonstrated a strong correlation between dysbiosis, defined as disturbances in the composition and/or functional characteristics of the microbiome, and the progression of colorectal cancer (CRC) [21]. In a recent study, Kadosh et al. demonstrated that dysbiosis can mediate the functional impairment of mutant p53 to promote the formation of intestinal tumors [22]. In contrast, p53 has been demonstrated to play a pivotal role in the suppression of the inflammatory microenvironment and the maintenance of intestinal epithelial integrity. Schwitalla et al. have demonstrated that the absence of p53 in mouse intestinal epithelial cells results in impaired integrity of the epithelial barrier, promoted bacterial infiltration, and widespread inflammation [23]. In the digestive system, p53 has been shown to regulate the gut microbiota in a beneficial manner by upregulating Mbl2 and Lcn2. This process enables p53 to play a role in protecting the intestinal epithelial barrier and preventing inflammation [24]. The dynamic interplay between p53 and gut microbiota represents a pivotal frontier in understanding cancer development. This rapidly evolving field not only reveals how dysbiosis and p53 dysfunction jointly fuel tumor progression but also opens promising avenues for microbiome-targeted therapeutic interventions against cancer.
Reference
- Alfason, L.; Li, W.; Altaf, F.; Wu, S.; Kasim, V. Resuscitating the Guardian: Current Progress in p53-Based Anti-Tumor Therapy. Onco Therapeutics 2021, 8, 51-92, doi:10.1615/ForumImmunDisTher.2021039201.
- Silva, J.L.; Lima, C.G.S.; Rangel, L.P.; Ferretti, G.D.S.; Pauli, F.P.; Ribeiro, R.C.B.; da Silva, T.B.; da Silva, F.C.; Ferreira, V.F. Recent Synthetic Approaches towards Small Molecule Reactivators of p53. Biomolecules 2020, 10, 635, doi:10.3390/biom10040635.
- Ceder, S.; Eriksson, S.E.; Liang, Y.Y.; Cheteh, E.H.; Zhang, S.M.; Fujihara, K.M.; Bianchi, J.; Bykov, V.J.N.; Abrahmsen, L.; Clemons, N.J.; et al. Correction: Mutant p53-reactivating compound APR-246 synergizes with asparaginase in inducing growth suppression in acute lymphoblastic leukemia cells. Cell Death Dis 2022, 13, 672, doi:10.1038/s41419-022-05130-y.
- Synnott, N.C.; O'Connell, D.; Crown, J.; Duffy, M.J. COTI-2 reactivates mutant p53 and inhibits growth of triple-negative breast cancer cells. Breast Cancer Res Treat 2020, 179, 47-56, doi:10.1007/s10549-019-05435-1.
- Rippin, T.M.; Bykov, V.J.; Freund, S.M.; Selivanova, G.; Wiman, K.G.; Fersht, A.R. Characterization of the p53-rescue drug CP-31398 in vitro and in living cells. Oncogene 2002, 21, 2119-2129, doi:10.1038/sj.onc.1205362.
- Peng, Y.; Bai, J.; Li, W.; Su, Z.; Cheng, X. Advancements in p53-Based Anti-Tumor Gene Therapy Research. Molecules 2024, 29, 5315, doi:10.3390/molecules29225315.
- Zhang, W.W.; Li, L.; Li, D.; Liu, J.; Li, X.; Li, W.; Xu, X.; Zhang, M.J.; Chandler, L.A.; Lin, H.; et al. The First Approved Gene Therapy Product for Cancer Ad-p53 (Gendicine): 12 Years in the Clinic. Hum Gene Ther 2018, 29, 160-179, doi:10.1089/hum.2017.218.
- Qi, L.; Li, G.; Li, P.; Wang, H.; Fang, X.; He, T.; Li, J. Twenty years of Gendicine(R) rAd-p53 cancer gene therapy: The first-in-class human cancer gene therapy in the era of personalized oncology. Genes Dis 2024, 11, 101155, doi:10.1016/j.gendis.2023.101155.
- Zhu, G.; Pan, C.; Bei, J.X.; Li, B.; Liang, C.; Xu, Y.; Fu, X. Mutant p53 in Cancer Progression and Targeted Therapies. Front Oncol 2020, 10, 595187, doi:10.3389/fonc.2020.595187.
- Batir, M.B.; Sahin, E.; Cam, F.S. Evaluation of the CRISPR/Cas9 directed mutant TP53 gene repairing effect in human prostate cancer cell line PC-3. Mol Biol Rep 2019, 46, 6471-6484, doi:10.1007/s11033-019-05093-y.
- Wang, J.; Liu, W.; Zhang, L.; Zhang, J. Targeting mutant p53 stabilization for cancer therapy. Front Pharmacol 2023, 14, 1215995, doi:10.3389/fphar.2023.1215995.
- Li, D.; Marchenko, N.D.; Moll, U.M. SAHA shows preferential cytotoxicity in mutant p53 cancer cells by destabilizing mutant p53 through inhibition of the HDAC6-Hsp90 chaperone axis. Cell Death Differ 2011, 18, 1904-1913, doi:10.1038/cdd.2011.71.
- Parrales, A.; Ranjan, A.; Iyer, S.V.; Padhye, S.; Weir, S.J.; Roy, A.; Iwakuma, T. DNAJA1 controls the fate of misfolded mutant p53 through the mevalonate pathway. Nature cell biology 2016, 18, 1233-1243, doi:10.1038/ncb3427.
- Tong, X.; Xu, D.; Mishra, R.K.; Jones, R.D.; Sun, L.; Schiltz, G.E.; Liao, J.; Yang, G.Y. Identification of a druggable protein-protein interaction site between mutant p53 and its stabilizing chaperone DNAJA1. J Biol Chem 2021, 296, 100098, doi:10.1074/jbc.RA120.014749.
- Polonsky, K.S. The past 200 years in diabetes. N Engl J Med 2012, 367, 1332-1340, doi:10.1056/NEJMra1110560.
- Vigneri, R.; Sciacca, L.; Vigneri, P. Rethinking the Relationship between Insulin and Cancer. Trends Endocrinol Metab 2020, 31, 551-560, doi:10.1016/j.tem.2020.05.004.
- Strycharz, J.; Drzewoski, J.; Szemraj, J.; Sliwinska, A. Is p53 Involved in Tissue-Specific Insulin Resistance Formation? Oxid Med Cell Longev 2017, 2017, 9270549, doi:10.1155/2017/9270549.
- Yan, Z.; Miao, X.; Zhang, B.; Xie, J. p53 as a double-edged sword in the progression of non-alcoholic fatty liver disease. Life Sci 2018, 215, 64-72, doi:10.1016/j.lfs.2018.10.051.
- Kung, C.P.; Murphy, M.E. The role of the p53 tumor suppressor in metabolism and diabetes. J Endocrinol 2016, 231, R61-R75, doi:10.1530/JOE-16-0324.
- Yahagi, N.; Shimano, H.; Matsuzaka, T.; Najima, Y.; Sekiya, M.; Nakagawa, Y.; Ide, T.; Tomita, S.; Okazaki, H.; Tamura, Y.; et al. p53 Activation in adipocytes of obese mice. J Biol Chem 2003, 278, 25395-25400, doi:10.1074/jbc.M302364200.
- Lau, H.C.H.; Yu, J. Gut microbiome alters functions of mutant p53 to promote tumorigenesis. Signal Transduct Target Ther 2020, 5, 232, doi:10.1038/s41392-020-00336-y.
- Kadosh, E.; Snir-Alkalay, I.; Venkatachalam, A.; May, S.; Lasry, A.; Elyada, E.; Zinger, A.; Shaham, M.; Vaalani, G.; Mernberger, M.; et al. The gut microbiome switches mutant p53 from tumour-suppressive to oncogenic. Nature 2020, 586, 133-138, doi:10.1038/s41586-020-2541-0.
- Schwitalla, S.; Ziegler, P.K.; Horst, D.; Becker, V.; Kerle, I.; Begus-Nahrmann, Y.; Lechel, A.; Rudolph, K.L.; Langer, R.; Slotta-Huspenina, J.; et al. Loss of p53 in enterocytes generates an inflammatory microenvironment enabling invasion and lymph node metastasis of carcinogen-induced colorectal tumors. Cancer Cell 2013, 23, 93-106, doi:10.1016/j.ccr.2012.11.014.
- Khor, A.H.P.; Koguchi, T.; Liu, H.; Kakuta, M.; Matsubara, D.; Wen, R.; Sagiya, Y.; Imoto, S.; Nakagawa, H.; Matsuda, K.; et al. Regulation of the innate immune response and gut microbiome by p53. Cancer Sci 2024, 115, 184-196, doi:10.1111/cas.15991.

Round 2
Reviewer 2 Report
Comments and Suggestions for Authors
Revised version of the reviewer article is acceptable.
Reviewer 3 Report
Comments and Suggestions for Authors
The authors answered my questions and comments. No more comments.